

# Viscosity of aqueous ammonium nitrate–organic particles: Equilibrium partitioning may be a reasonable assumption for most tropospheric conditions

Liviana K. Klein[1,*], Allan K. Bertram[2], Andreas Zuend[3], Florence Gregson[2], and Ulrich K. Krieger[1]

[1]Institute for Atmospheric and Climate Science, ETH Zurich, Zurich, Switzerland
[2]Department of Chemistry, University of British Columbia, Canada
[3]Department of Atmospheric and Oceanic Sciences, McGill University, Montreal, Quebec, Canada
[*]Now at: Department of Civil and Environmental Engineering, Virginia Tech, Blacksburg, USA

**Correspondence:** Liviana Klein (livianaklein@vt.edu), Ulrich Krieger (ulrich.krieger@env.ethz.ch)

**Abstract.**

The viscosity of aerosol particles determines the critical mixing time of gas–particle partitioning of volatile compounds in the atmosphere. The partitioning of the semi-volatile ammonium nitrate ($NH_4NO_3$) might alter the viscosity of highly viscous secondary organic aerosol particles during their lifetime. In contrast to the viscosity of organic particles, data on the viscosity of

internally mixed inorganic–organic aerosol particles are scarce. We determined the viscosity of an aqueous ternary inorganic–organic system consisting of $NH_4NO_3$ and a proxy compound for a highly viscous organic, sucrose. Three techniques were applied to cover the atmospherically relevant humidity range: viscometry, fluorescence recovery after photobleaching, and the poke-flow technique. We show that the viscosities of $NH_4NO_3$–sucrose–$H_2O$ with an organic to inorganic dry mass ratio of 4:1 are four orders of magnitude lower than those of the aqueous sucrose under low humidity conditions (30% relative

humidity (RH), 293 K). By comparing viscosity predictions of mixing rules with those of the AIOMFAC-VISC model, we found that a mixing rule based on mole fractions performs similarly when data from corresponding binary aqueous subsystems are available. Applying this mixing rule, we estimated the characteristic internal mixing time of aerosol particles, indicating significantly faster mixing for inorganic—organic mixtures compared to electrolyte-free particles, especially at lower RH's. Hence, the assumption in global atmospheric chemistry models of quasi-instantaneous equilibrium gas–particle partitioning is

reasonable for internally mixed single-phase particles containing dissolved electrolytes (but not necessarily for phase-separated particles), for most conditions in the planetary boundary layer. This assumption may even hold for the entire troposphere at mid-latitudes and RH > 35%.





# 1   Introduction

Aerosol particles are emitted either directly by anthropogenic activities and natural processes or form through gas–particle
conversion. Their composition can be very complex, consisting of organic compounds, inorganic salts, or an internal mixture
of both (Kanakidou et al., 2005; Fuzzi et al., 2006; Riemer et al., 2019). During their atmospheric lifetime, the composition
may change via chemical reactions and gas–particle partitioning. The composition determines the physicochemical properties,
such as the viscosity. Since molecular diffusivity is inversely proportional to dynamic viscosity, it affects the equilibration time
scales of gas–particle partitioning of semivolatile organic and inorganic compounds, and consequently long-range pollutant
transport via aerosol particles (Shrivastava et al., 2017; Bastelberger et al., 2017; Mu et al., 2018). Global models often assume
equilibrium partitioning is achieved for fine particulate matter within the typical model time steps used for periodic output (tens
of minutes to ca. 1 hour, e.g. Bian et al. (2017)).

Ammonium nitrate ($NH_4NO_3$) is a semivolatile pollutant that is in equilibrium with ammonia and nitric acid in the gas phase
and dissociates into ammonium and nitrate ions ($HNO_3(g) + NH_3(g) \rightleftharpoons NO_3^-(aq) + NH_4^+(aq)$) in an aqueous particle phase.
Aqueous $NH_4NO_3$ generally remains in its ionic form and does not effloresce in aqueous solution at low relative humidities
(Dassios and Pandis, 1999). Efflorescence is unlikely to occur in internally mixed aerosol containing secondary organic matter
together with $NH_4NO_3$ (for example, Marcolli et al. (2004)). Currently, little is known about mass transfer limitations in
viscous mixed $NH_4NO_3$–organic particles, potentially impeding gas–particle partitioning of ammonia and nitric acid. Due
to significant efforts to reduce sulfur emissions in recent decades, the particulate nitrate/sulfate ratio has generally increased
(Weber et al., 2016). For example, following the implementation of the Clean Air Act in Beijing (2014 - 2017), the mass ratio
of particulate nitrate/sulfate increased from 1.5 to 3.7 (Li et al., 2019). Recently, Liu et al. (2023) showed with particle rebound
experiments that the relative humidity threshold for the transition from solid to liquid shifts to lower RH with increasing
inorganic compound mass fraction. They concluded: "This allowed urban aerosol particles to exist in liquid state at lower RH,
and consequently, kinetic limitation by bulk diffusion in nitrate-dominated particles might be negligible,...". This motivated us
to study the viscosity of mixed $NH_4NO_3$–organic particles in detail in this work.

Only a few laboratory studies have been reported investigating the viscosity of internally mixed inorganic–organic aerosol
systems (Tong et al., 2022; Song et al., 2021; Jeong et al., 2022; Rovelli et al., 2019; Power et al., 2013; Richards et al., 2020;
Sheldon et al., 2023). The addition of inorganic salts to organic material in a liquid and homogeneously mixed particle is
expected to decrease viscosity by orders of magnitude for all water activties for monovalent ions (Tong et al., 2022; Rovelli
et al., 2019; Jeong et al., 2022). This decrease in viscosity is related to the plasticizing effect of the water associated with the
hygroscopic salts. In contrast to these findings, Richards et al. (2020) and most recently Sheldon et al. (2023), report increased
viscosities for certain mixed inorganic–organic systems compared to aqueous organic ones. Gel formation may explain this
behavior for systems containing divalent ions (e.g. $Ca^{2+}$ and $Mg^{2+}$) as suggested by Richards et al. (2020). We know of only
one study investigating the viscosity of an $NH_4NO_3$–organic system: Tong et al. (2022) reported an approximately three orders
of magnitude lower viscosity at 40% RH for levitated $NH_4NO_3$–sucrose particles with an organic to inorganic dry mass ratio
(OIR) of 1 at a temperature of 298 K compared to binary aqueous sucrose particles under the same conditions.





Currently, there are several methods to predict the viscosity of ternary systems: mixing rules based on data from the corresponding binary, aqueous mixtures, and the AIOMFAC-VISC thermodynamic model, which is applicable to any number of components of the mixture (Lilek and Zuend, 2022). Mixing rules have already been successfully applied to predict the

viscosity of binary systems from data from pure components (Song et al., 2016b; Gervasi et al., 2020). However, for ternary systems, mixing rules have been tested in only two studies (Mahant et al., 2023; Rovelli et al., 2019). Of these two, only Rovelli et al. (2019) applied them to a mixed inorganic–organic system. They concluded that a mixing rule following the Zdanovskii-Stokes-Robinson (ZSR) relationship can predict their system reasonably well. However, these and other simple mixing rules can only be applied if the viscosity of the binary systems is known. On the contrary, thermodynamic models such as Aerosol

Inorganic–Organic Mixtures Functional groups Activity Coefficients Viscosity model (AIOMFAC-VISC) can predict the viscosity of complex mixtures of many components. However, data to validate and potentially improve this model are scarce. So far, it has only been tested for a limited amount of inorganic–organic sucrose mixtures with $NaNO_3$, $(NH_4)_2SO_4$, $Ca(NO_3)_2$, and $Mg(NO_3)_2$ (Lilek and Zuend, 2022).

To address the shortcomings outlined above, we investigated internally mixed aqueous inorganic–organic solutions, droplets,

and films consisting of $NH_4NO_3$ and a proxy for highly viscous organics, sucrose. We applied three different experimental techniques to determine viscosity in the atmospherically relevant RH range at room temperature ($293.15 \pm 1$ K) for OIRs 1 and 4, namely viscometry, rectangular fluorescence recovery after photobleaching (rFRAP) and the poke flow technique. We applied three different simple mixing rules to predict the viscosities of the ternary solutions based on those of the aqueous binary solutions, carried out simulations with AIOMFAC-VISC, and evaluated the performances for all. Furthermore, published data

on the viscosity of inorganic–organic mixed aerosol particles were collected, and we also assessed the applicability of the three mixing rules in this modest dataset. Finally, we used the average compositional data of aerosol particles collected by Zhang et al. (2011) to estimate the viscosities and the corresponding characteristic mixing times of internally mixed single phased aerosol particles at three different locations to evaluate whether equilibrium partitioning is a valid assumption in global models for typical conditions of the planetary boundary layer (PBL).

## 2    Methods

### 2.1    Sample Preparation

For all viscosity measurements, bulk solutions of $NH_4NO_3$–sucrose–$H_2O$ were prepared beforehand. Sucrose (purity 99%, Alfa Aesar) and $NH_4NO_3$ (purity $\geq$ 99%, Sigma-Aldrich) were used. Solutions containing dry organic to inorganic ratios (OIR) of 1 and 4 were prepared at 20 wt% with Millipore-water (resistivity $\geq$ 18.2 $M\Omega\,cm$). The solutions were freshly

prepared before each measurement to avoid evaporation of $HNO_3$ and $NH_3$ from the solution before the experiment.





## 2.2 Poke-Flow Technique to Measure Viscosity in Droplets

The poke-flow technique has been described in detail elsewhere (Murray et al., 2012; Renbaum-Wolff et al., 2013b; Grayson et al., 2015). To generate sample droplets, the solution was nebulized onto hydrophobic slides (d = 11 mm, Hampton Research, Canada) obtaining droplets with diameters of 50 - 200 μm. Subsequently, the slides were placed in a flow cell and mounted

above an objective (40x) of an inverse optical microscope (AmScope, ME1400TC-INF). The flow cell was maintained at room temperature (293 ± 1 K) and relative humidity was controlled. The RH was controlled by combining dry and humidified zero air with flow rates in the range of 500 - 1000 sccm. The relative humidity is determined via temperature (measured with a thermocouple mounted on the flow cell, OMEGA, HH-200a) and the dew point temperature measured with a chilled mirror hygrometer (General Eastern, optical dew point sensor, 1311DR, uncertainty ± 2.5%) downstream of the cell. After RH was

adjusted, we allowed the droplets to equilibrate with the surrounding conditions (see SI for details). After conditioning, the droplets were poked with a 25 μm diameter tungsten needle (Roboz Surgical Instruments Co.). This deformed the droplet into a half-torus geometry. To minimize surface energy, the droplets recovered to a minimum surface (near-spherical) shape. The time needed to recover, the experimental relaxation flow time, was measured by taking pictures with a camera (AmScope, MA1000) every 500 ms. Pictures were analyzed with the software ImageJ (Collins, 2007). The outer area of the particle and

the area of the poked hole were determined right after poking and after a certain relaxation time, namely, until the hole created by poking has reached 1/4 of its initial area or less if the relaxation time is in the order of days. Figure 1 shows a typical poke-flow experiment. The relaxation time was used to obtain the viscosity of this process through fluid dynamics simulations with the COMSOL multiphysics software, details are explained in Grayson et al. (2015). Surface tension, slip length, and contact angle are input parameters for the simulation and are shown in Table 1. The contact angle of a $NH_4NO_3$–sucrose–$H_2O$ (OIR

= 4) droplet was measured with a laser scanning microscope (see Fig. A1). Slip length and surface tension values were based on measurements reported in the literature and represent conservative upper and lower limits (see Table 1, footnotes). At least three replicates were performed for each RH. The maximum upper and minimum lower limits of the viscosities are reported as error bars in Figures 4 and 5 in the results section. If the droplets cracked upon poking, we report a lower limit of $5 \cdot 10^8$ Pa s based on reference experiments by Renbaum-Wolff et al. (2013a).

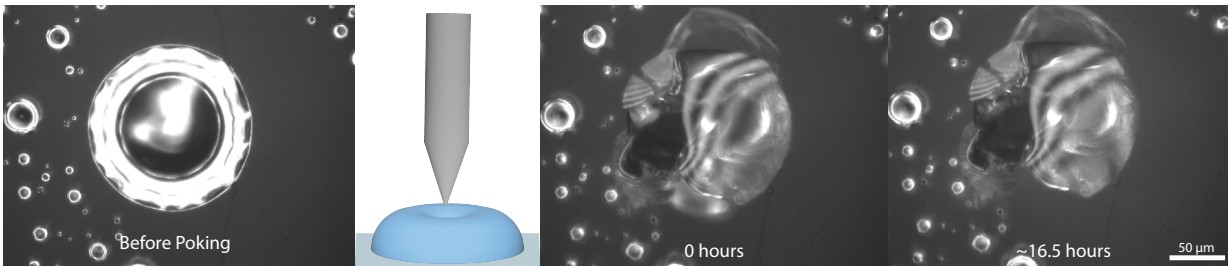

**Figure 1. Example of a poke-flow experiment.** Microscopy images (40x) taken during the experiment before and after poking of a sucrose–$NH_4NO_3$–$H_2O$ (OIR = 4) droplet at 5% RH and 293 ± 1 K. The droplet cracks during poking and does not show any movement for approximately 16 hours.



**Table 1.** Parameters for the COMSOL flow simulation for $NH_4NO_3$–sucrose–$H_2O$ (OIR = 4) experiments at $293 \pm 1$ K.

|                       | Surface Tension (m $Nm^{-1}$) | Slip length (m)       | Contact Angle (°) |
| --------------------- | ----------------------------- | --------------------- | ----------------- |
| Lower limit viscosity | $57.2^a$                      | $5 \cdot 10^{-9,c}$   | $78.44^d$         |
| Upper limit viscosity | $75.6^b$                      | $1 \cdot 10^{-5,c}$   | $69.1^d$          |

[a] This value has been used previously to represent a lower limit for sucrose-water solutions (Grayson et al., 2015; Rovelli et al., 2019). This number is a also a lower limit to the surface tension observed in levitated sucrose-water droplets (Power, 2014; Singh et al., 2023).

[b] The surface tension of sucrose based on the extrapolation of aqueous sucrose measurements (MacDonald et al., 1996).

[c] The slip length are based on measurements of water and organic compounds on hydrophobic surfaces (see references in Grayson et al. (2015))

[d] To obtain a lower limit of the contact angle, Rhodamine 6G was added to the bulk solution and nebulized onto a siliconized glass slide (Hampton Research, Canada) at room conditions. The slide was placed under a confocal laser scanning fluorescence microscope (Zeiss Axio Observer LSM 510MP, oil objective, 60x magnification). A vertical profile was recorded for more than three droplets. With the height and radius of the vertical profile, the upper and lower limit of the contact angle were calculated, assuming a spherical geometry. Note, the simulated viscosities only depended weakly on the contact angles. For example, if the contact angle is changed by $\pm$ 10 % the simulated viscosities change on average by only roughly $\pm$ 15 %, which is small compared to the overall uncertainties in the simulated viscosities (> 10x) (Rovelli et al., 2019).

## 2.3   rFRAP to Determine Viscosities

The rectangular fluorescence recovery after photobleaching method (rFRAP) was previously described in detail (Deschout et al., 2010; Evoy et al., 2021; Chenyakin et al., 2017). With this method, the diffusion coefficient of a fluorescent dye that diffuses through a sample solution can be determined. In short, a rectangular square is photobleached in the sample and then after some time fluorescence in this photobleached region recovers, due to diffusion of unbleached dye into the photobleached region, and photobleached dye molecules out into the unbleached region. From the time dependence of this recovery, a diffusion coefficient, $D_{dye}$, of the dye in the sample solution was determined. Afterward, the $D_{dye}$ was used to calculate the dynamic viscosity, $\eta$, of the mixture, assuming that the Stokes–Einstein relation holds:

$$D_{dye} = \frac{kT}{6\pi \eta r_{dye}}, \tag{1}$$

where $k$ is the Boltzmann constant, $T$ is the absolute temperature, and $r_{dye}$ is the apparent radius of the diffusing molecule, which is assumed to be spherical. Since the size of the fluorescent dye used in these experiments is similar to or larger than the matrix molecules, the Stokes–Einstein relation should give reasonable estimates of the viscosities in these experiments (Evoy et al., 2020, 2019; Chenyakin et al., 2017). To prepare the sample for the rFRAP experiments, first, Rhodamine B isothiocyanate dextran (Sigma-Aldrich, $r_{dye} = 5.8$ nm, (Paës et al., 2017)) was added as a fluorescent dye to the bulk solutions of aqueous $NH_4NO_3$–sucrose (0.01 mM). The solution was pipetted onto glass slides (Hampton Research, Canada), which were covered with vacuum grease on the perimeter. The samples were equilibrated above a saturated salt solution within a sealed glass jar, such that the headspace exhibited a controlled RH (at $293 \pm 1$ K, $K_2CO_3$, NaBr and NaCl for RH = 43%, 59% and 75%,



respectively). The time required for the droplets to equilibrate with the gas phase was estimated using the characteristic mixing time $\tau$ of a droplet of radius $r_{\mathrm{droplet}}$ (Preston, 2022)

$$\tau_{\mathrm{mix}}(T, a_w) = \frac{r_{\mathrm{droplet}}^2}{\pi^2 \cdot D_{\mathrm{H_2O}}(T, a_w)}, \tag{2}$$

where $D_{\mathrm{H_2O}}$ is the diffusion coefficient of water in the condensed phase. The determination of the diffusion coefficients for the solution species of interest is described in detail in the Appendix C. After equilibration, the jar containing the glass slides was placed inside a glove bag (Glas-Col, Terre Haute USA) with controllable RH. RH was controlled using a dry and humidified flow of nitrogen gas and is monitored with a handheld hygrometer (Omega RH85, uncertainty $\pm$ 2.5%). When the RH in the glove bag reached the same RH as that in the jar, the jar was opened. A second glass slide was placed on top of the first glass

slide containing the droplets, sandwiching the droplets into thin films ranging from 25-140 μm. Subsequently, the prepared sample was placed under a confocal laser scanning fluorescence microscope (Zeiss Axio Observer LSM 510MP, 0.3 NA objective, 10x) at room temperature (293 $\pm$ 1 K). A rectangular region was photo-bleached in the center of the sample using a 543 nm HeNe laser. The intensity of the laser was adjusted so that the fraction of photo-bleached molecules is around 30%. Next, the confocal laser scanning fluorescence microscope was used to record fluorescence images of the sample and monitor

the recovery of the fluorescence intensity in the photo-bleached region. Fluorescence images were processed with the software ImageJ (Collins, 2007). A time-lapse of a representative rFRAP experiment is shown in Figure 2. A Matlab software program was used to determine the diffusion coefficient of the processed images using a mathematical description of the fluorescent intensity as a function of position and time after photobleaching as described in Deschout et al. (2010).

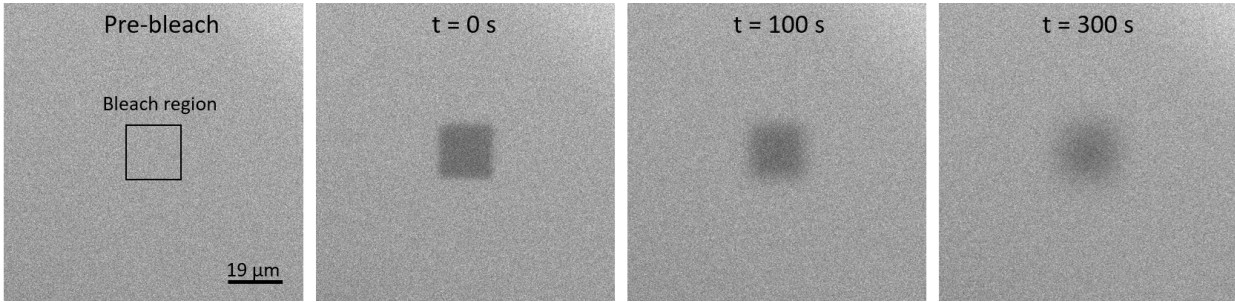

**Figure 2. Images from a typical rFRAP experiment.** Diffusion of Rhodamine B isothiocyanate dextran in $NH_4NO_3$–sucrose–$H_2O$ (OIR = 4) at 59% RH and 293 K. The first image shows the sample prior to photobleaching with the black square indicating the region to be photobleached (19 μm x 19 μm). The other three images show the sample at different time points after photobleaching (t = 0 s, 100 s, and 300 s) and the recovery of the fluorescence.

## 2.4 Viscometry to Measure Viscosity in Bulk Solutions

For viscometer experiments, bulk solutions near saturation were prepared for OIR 1 and 4 and diluted with Milli-Q water to reach higher water activities. Water activities were measured for these compositions with an Aqualab water activity meter ($a_w$ precision of $\pm$ 0.003). The viscosity of the sample solution was measured on the day of preparation to avoid evaporation. Since



these were independent measurements, we approximated the uncertainty of water activity with $\pm$ 2% to account for small compositional differences.

Most bulk viscosity measurements were performed by the UBC BioProducts Institute with a rheometer (MCR302, Anton-Parr). Three replicates were performed for each water activity at a temperature of $298.1 \pm 0.1$ K. The shear rate varied from 1 to $1000\,\mathrm{s}^{-1}$. The viscosity average over all shear rates is reported for each solution assuming a Newtonian fluid.

The viscosities obtained for 298 K were converted to other temperatures, e.g. 293 K, by application of the Arrhenius equation for better comparability with our other measurements, which are conducted at 293 K,

$$\ln\left(\frac{\eta}{\eta_{\mathrm{ref}}}\right) = -\frac{E_A}{R}\left(\frac{1}{T_{\mathrm{ref}}} - \frac{1}{T}\right).\tag{3}$$

Here, $E_A$ is the apparent activation energy of diffusion for viscous flow; for dilute aqueous sucrose solutions $E_A$ is estimated to be $50\,\mathrm{kJ\,mol}^{-1}$ (Kiland et al., 2019). $\eta$ is the dynamic mixture viscosity at temperature $T$, $\eta_{\mathrm{ref}}$ is the corresponding viscosity at the reference temperature $T_{\mathrm{ref}}$ and $R$ is the ideal gas constant.

Two additional viscosities for OIR=1 at $a_w = 0.68$ and 0.73 were measured with the Viscometer LVDV-II+ (Brookfield) with
Spindle Yula-15 (1–2000 mPa s). The water bath was set to $T = 293$ K and the sample was equilibrated with the water bath for at least 15 min. Three replicates were performed for each water activity. The water activity was measured with Aqualab for the same solutions.

## 2.5 Mixing Rules to Predict Ternary Mixture Viscosities

To predict the viscosity of ternary solutions from known viscosities of binary aqueous solutions, we applied three different
mixing rules. They all follow the same mathematical principle of summing the logarithms of the viscosities of the binary solutions, but with different meanings of the coefficients $f_1$ and $f_2$:

$$\log_{10}\left(\frac{\eta(a_w)}{\eta_0}\right) = f_1 \log_{10}\left(\frac{\eta_1(a_w)}{\eta_0}\right) + f_2 \log_{10}\left(\frac{\eta_2(a_w)}{\eta_0}\right),\tag{4}$$

where $\eta$ is the viscosity of the ternary solution at a particular water activity, $\eta_0$ denotes unity viscosity (1 Pa s), $\eta_i(a_w)$ are the viscosities of the binary aqueous solutions at the same water activity.

First, we use a mole-fraction-based mixing rule, which was proposed by Arrhenius (1887) for predicting the viscosity of binary solutions, where $f_1$ and $f_2$ are the dry mole fractions of the components 1 and 2. Second, the same approach is used but mole fractions are replaced by mass fractions ($w$). Finally, and alternatively, Zdanovskii–Stokes–Robinson (ZSR) (Zdanovskii, 1948; Stokes and Robinson, 1966) proposed an approach for estimating the water content of ternary solutions by adding up the water contents of the two corresponding binary subsystems at the same water activity. A similar approach can be used to estimate the
viscosity of a ternary mixture based on the corresponding binary aqueous subsystem viscosities at given RH (or water activity). We follow the same approach as applied by Song et al. (2021) and described in detail by Lilek and Zuend (2022): where $f_1$ is

$$f_1 = \frac{\mathrm{OIR} \cdot w_{\mathrm{el}}(a_w)}{w_{\mathrm{sucrose}}(a_w) + \mathrm{OIR} \cdot w_{\mathrm{el}}(a_w)}\tag{5}$$



and $f_2 = 1 - f_1$, $w$ is the mass fraction and OIR is the organic-to-inorganic dry mass ratio.

The ZSR-based method preserves the OIR of the original multicomponent mixture, while the mole-fraction-based mixing rule
does not. In addition, the weighting factors ($f_1$, $f_2$) need to be recomputed for each water activity level, since they are not
constant even when the OIR is constant (Lilek and Zuend, 2022).

In this study, we applied the outlined viscosity mixing rules to the $NH_4NO_3$–sucrose–$H_2O$ systems that we studied in detail,
but also to published data for inorganic–organic mixed particles containing water, sucrose and one of the following salts:
$NaNO_3$, $NaCl$, $(NH_4)_2SO_4$, $Ca(NO_3)_2$ or $Mg(NO_3)_2$ (Rovelli et al., 2019; Power et al., 2013; Tong et al., 2022; Jeong et al.,
2022; Song et al., 2021).

Viscosities for aqueous inorganics were obtained from the CRC Handbook (Rumble, 2023), Laliberté collection (Laliberté,
2007) and other published data (Power et al., 2013; Song et al., 2021; Baldelli et al., 2016). Equilibrium water activities were
calculated for the binary concentrations using the thermodynamic models E-AIM and AIOMFAC at 293.15 K, except where
otherwise indicated. We use E-AIM for $NH_4NO_3$, $(NH_4)_2SO_4$, $NaNO_3$ (298.15 K), and $NaCl$ (298.15 K) (Clegg et al., 1998;
Friese and Ebel, 2010)) and since $Ca(NO_3)_2$ and $Mg(NO_3)_2$ are not parameterized in E-AIM we use AIOMFAC instead
(Zuend et al., 2008, 2011). We fitted polynomial functions to these binary aqueous–salt data to parameterize the viscosity as
a function of water activity. For $NH_4NO_3$ we only considered the data by Laliberté (Laliberté, 2007) since it was measured
at the temperature of interest and also added a data point for the viscosity under dry conditions estimated from the subcooled
$NH_4NO_3$ melt viscosity (Booth and Vinyard, 1967) as explained in the Appendix A (see Fig. 3A). For aqueous sucrose, we
fitted a polynomial function to the previously published viscosity data, our measurements, and assumed a viscosity of $10^{12}$ Pa
for the glass transition at 28% RH and 293 K (Zobrist et al., 2008) (see Fig. 3B). The parameters for all fits are given in Table
A1.





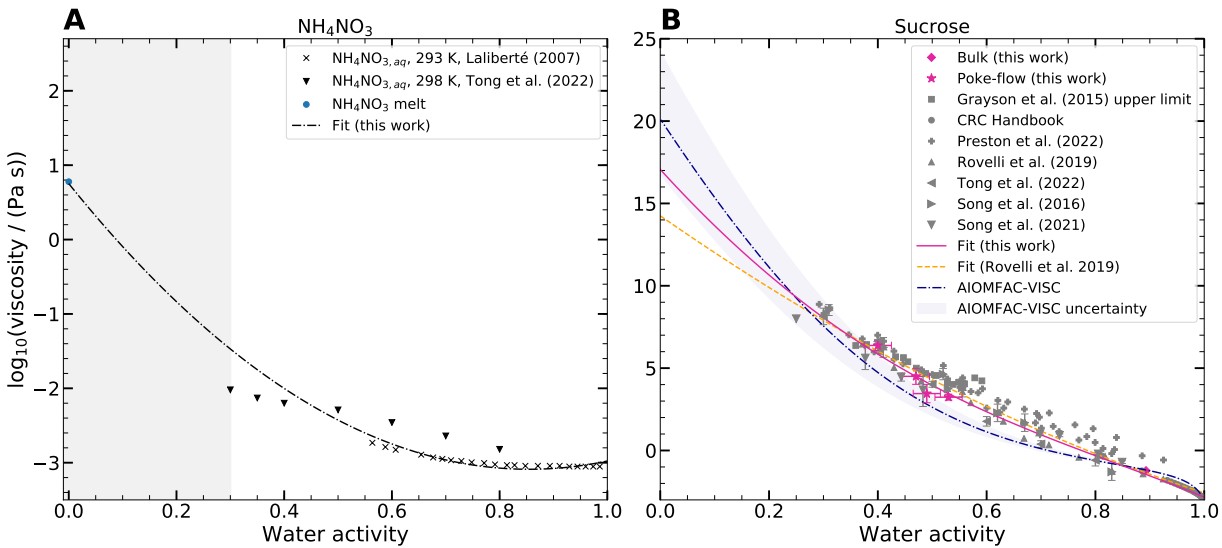

**Figure 3. Measured and predicted $\log_{10}$(viscosity/(Pa·s)) of binary mixtures as a function of water activity at 293–298 K.** Panel **(A)** shows $\log_{10}$(viscosity/(Pa·s)) of aqueous $NH_4NO_3$ from published data (Laliberté, 2007; Tong et al., 2018; Booth and Vinyard, 1967) and a polynomial fit of the Laliberté and Booth data (gray-dashed line). Panel **(B)** shows $\log_{10}$(viscosity/(Pa·s)) of aqueous sucrose measured in this work and published experimental data, a polynomial fit (pink solid line), a fit by Rovelli et al. (2019) (orange dashed line) and independent predictions by the AIOMFAC-VISC model (blue dash dotted line).

## 2.6 AIOMFAC-VISC to Predict Multi-Component Mixture Viscosities

AIOMFAC-VISC is an extension of the AIOMFAC model (Zuend et al., 2008, 2011) for the purpose of thermodynamics-based
viscosity predictions in liquid (amorphous) phases. This viscosity model is described in detail by Gervasi et al. (2020) and Lilek and Zuend (2022). AIOMFAC is a thermodynamic group-contribution model primarily for the prediction of non-ideal mixing effects in multicomponent solutions, e.g. to determine activity coefficients and to facilitate the prediction of liquid–liquid phase separation under given equilibrium conditions. With the update by Lilek and Zuend (2022), AIOMFAC-VISC can predict the viscosity of organic–inorganic aerosol particle phases as a function of composition and temperature in addition to other outputs
such as water activity. Pure-component viscosity values of organic mixture components at a temperature of interest need to be provided as input to AIOMFAC-VISC or, alternatively, can be estimated during calculations via parameterizations or structure–activity relationships (e.g., DeRieux et al., 2018; Armeli et al., 2023).

The viscosity of $NH_4NO_3$ was modeled with the online version of the AIOMFAC-VISC model, while sucrose was modeled with an updated offline version (Jeong et al., 2022) to capture the water uptake of sucrose more accurately. For ternary solutions,
the so-called "aquelec mixing" model was applied, in which the viscosity of the solution was computed using the modified viscosity of water properties in an aqueous organic subsystem. This is achieved by accounting for how the electrolytes dissolved in the available water impact the viscosity of this hypothetical organic-free subsystem (see Lilek and Zuend (2022) for details).




In this work, predictions were made for the same systems and OIR conditions as those studied in the experiments. All viscosities are predicted for 293.15 K.

## 3 Results and Discussion

### 3.1 Viscosity Measurements (Viscometry, Poke-Flow, and rFRAP)

In order to determine the viscosities of the ternary mixture of $NH_4NO_3$–sucrose–$H_2O$ across the full water activity range, we applied three different measurement techniques: viscometry, the poke-flow technique, and rFRAP. In this study, we measured mixtures of $NH_4NO_3$–sucrose–$H_2O$ with OIRs of 1 and 4, which is similar to what has been observed in terms of organic-to-inorganic mass ratios in the lower troposphere (Zhang et al., 2011).

The viscosity data of all measurements are shown in Fig. 4 (viscometry (circles), poke-flow (diamonds), and rFRAP (square)), together with the data from Tong et al. (2022). We provide regression lines for aqueous sucrose and aqueous $NH_4NO_3$ for comparison (gray dashed line and gray dotted-dashed line, respectively).

For the bulk solutions we determined viscosities from 0.008 Pa s to 0.017 Pa s ($a_w$ = 0.74 to $a_w$ = 0.635) for an OIR of 1 (blue) and 0.003 to 0.05 Pa s ($a_w$ = 0.93 to $a_w$ = 0.76) for an OIR of 4 (pink). For comparison, all of these solutions are less viscous than $10^{-1}$ Pa s (typical of olive oil, Koop et al. (2011)).

The viscosities obtained via FRAP (squares, $a_w$ = 0.75 to $a_w$ = 0.43) agree well with the bulk measurements and continue the trend towards higher viscosities. The viscosity of the solution with OIR = 1 with a water activity of 0.43 is approximately $10^{-1}$ Pa s (similar to olive oil) and the one for the OIR = 4 is $10^2$ Pa s (similar to ketchup, Koop et al. (2011)).

With the poke-flow technique, for OIR = 4 we could obtain viscosities in the water activity range from 0.25 to 0.05. Viscosities range from $10^3$ Pa s (peanut butter) to $10^8$ Pa s (pitch). The droplets at $a_w$ of 0.05 cracked upon poking; thus a lower limit of $5 \cdot 10^8$ Pa s was assumed. For the OIR = 1 sample, only an upper and lower limit (arrows) of the viscosity could be obtained. The droplets recovered immediately after the needle was removed for water activities $\geq 0.1$. Since the shape recovery was faster than the fastest recovery of the sample with OIR = 4, we assume that the viscosity measured from the 4 sample is an upper limit for the sample with OIR = 1 ($\eta \leq 10^3$ Pa s). The droplet cracked when poked for water activities $\leq 0.05$, i.e. $\eta \geq 5 \cdot 10^8$ Pa s.

The measured viscosities cover a wide range of about 10 orders of magnitude ($10^{-1}$ Pa s to $10^8$ Pa s). We observed a very sharp transition from liquid to semi-solid/glassy under low humidity conditions for OIR = 1 (between $a_w$ of 0.1 and 0.05) and a more gradual increase for OIR = 4 (between 0.25 and 0.05). The viscosities of the mixed systems are between the viscosities of the binary solution of aqueous $NH_4NO_3$ and aqueous sucrose. Furthermore, the viscosities of OIR = 1 are lower than those of OIR = 4. A decrease in viscosity is expected when hygroscopic ions are added to an organic matrix due to the plasticizing effect of water (Zobrist et al., 2008; Koop et al., 2011).



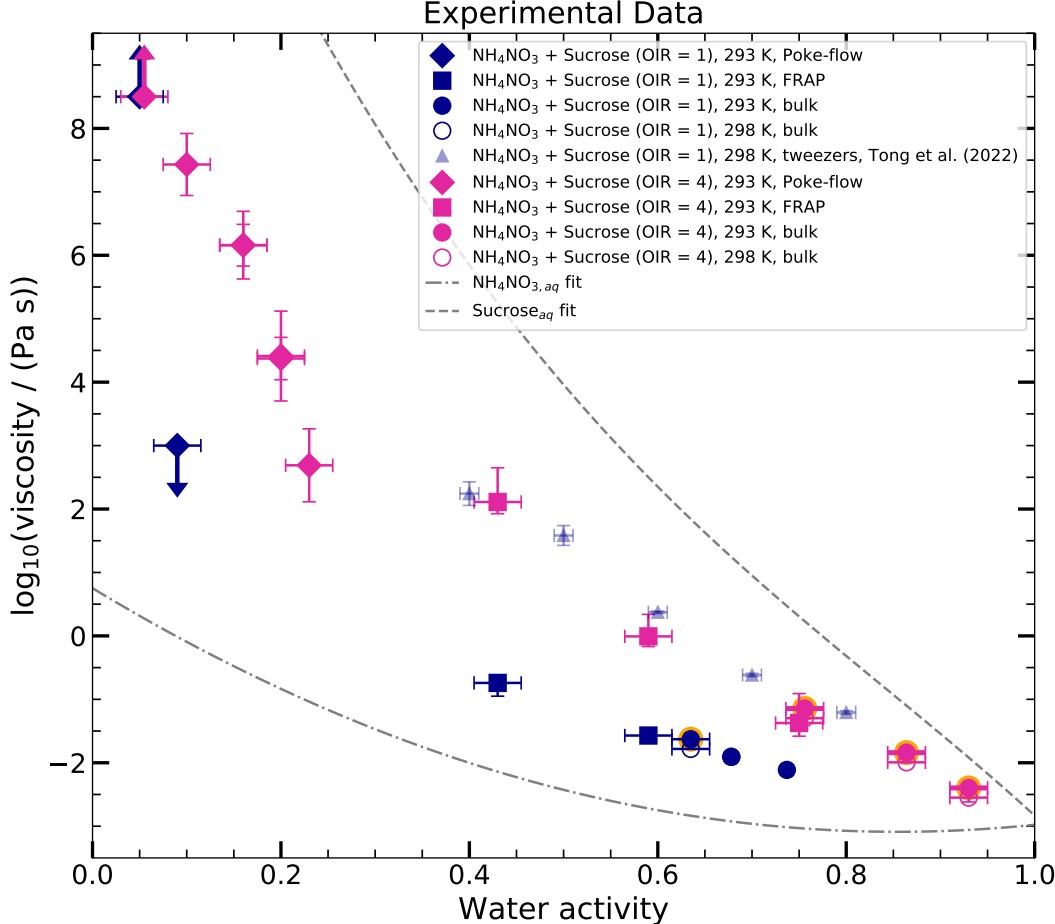

**Figure 4. Experimental data of the log$_{10}$(viscosity/(Pa· s)) as a function of water activity of the ternary system sucrose–NH$_4$NO$_3$–H$_2$O with OIR = 1 (blue) and OIR = 4 (pink) at 293 K and 298 K**. Different symbols indicate various measurement techniques: poke-flow (diamonds), rFRAP (squares), and viscometry/bulk (circles). The results of the poke-flow measurements are shown as the maximum and minimum of the simulated upper and lower viscosities of at least three replicates for each water activity. Diamonds represent the midpoints to guide the eye. Arrows in the upward and downward directions represent the lower and upper limits of the viscosities, respectively. Several error bars are present at the same water activity if two measurements were conducted for different equilibration times. The uncertainty in a$_w$ for the poke-flow technique originates from the uncertainty of the hygrometer, ± 2.5%. The rFRAP results (squares) represent the mean of at least three repeat measurements, with the error bar representing the maximum and minimum of the repeats. The a$_w$-error for the rFRAP results from the uncertainty of the hygrometer ± 2.5%. Open circles indicate bulk measurements at 298 K that are converted to 293 K (filled circles with orange edges) using the Arrhenius equation. Bulk measurements at 293 K are shown with filled circles without orange edges. The a$_w$-error is assumed to be ± 2% due to possible compositional differences between the sample used for the viscosity measurement and the one used to measure the water activity. When the water activity was measured from the same sample as the one used in the viscosity measurements, the a$_w$-error is smaller than the data point. Fitted curves for aqueous sucrose (gray dashed line) and aqueous NH$_4$NO$_3$ (gray dotted-dashed line) are shown for comparison. The light blue triangles are published data for an OIR = 1 measured with optical tweezers by Tong et al. (2022).



The only existing data for aqueous $NH_4NO_3$–sucrose (OIR = 1) were reported by Tong et al. (2022) who used optical tweezers to measure viscosity. Their data are approximately one order of magnitude higher than ours for $a_w \approx 0.7$, and almost three orders of magnitude for $a_w \approx 0.4$. These differences significantly exceed the measurement uncertainties reported. Since the uncertainties of the bulk viscometry measurements are well characterized ($\pm$ 1% according to the manufacturer), we conclude that the Tong et al. (2022) data are questionable. A possible reason could be the volatilization of $NH_4NO_3$ during their experiment, which is consistent with their viscosities being shifted to higher values.

The trend of viscosities with water activity observed in this study is similar to previously reported data for sucrose mixtures with other inorganic salts (Rovelli et al., 2019; Song et al., 2021; Jeong et al., 2022). As water acts as a plasticizer, a decrease in water activity is generally associated with an increase in viscosity. However, unlike Song et al. (2021) and the speculations of Jeong et al. (2022), who looked at different systems that more easily crystallize, in our system we do not believe that a phase change is responsible for the significant increase in viscosity below $a_w$ of 0.1. Our optical images indicate that $NH_4NO_3$ does not effloresce in our droplets, and a gel transition, as suggested by Song et al. (2021), is not expected for monovalent ions.

## 3.2 Viscosity Predictions of Ternary Solutions with Mixing Rules and AIOMFAC-VISC

Based on the viscosity measurements that we performed on the mixtures of $NH_4NO_3$–sucrose–$H_2O$ for two OIRs, we examined whether the viscosity of the ternary mixture could be predicted using different mixing rules and/or AIOMFAC-VISC. We studied three mixing rules: one based on mole fraction, one on mass fraction and another one based on the ZSR approach (Eq. 4). These rules were applied to the $\log_{10}$(viscosities/(Pa· s)) using our regressions for $\log_{10}$(viscosity/(Pa·s)) as a function of $a_w$ for binary mixtures of aqueous sucrose and aqueous $NH_4NO_3$, as shown in Figure 4. Additionally, the ternary mixtures were modeled using AIOMFAC-VISC.

Figure 5 shows the viscosities obtained for the ternary mixtures. The first column represents the results obtained using mixing rules, while the second column shows the results obtained using AIOMFAC-VISC. The upper row corresponds to OIR = 1, and the lower row corresponds to OIR = 4. To provide a comparison, the aqueous binary mixtures are also included in the figure, represented by a gray dashed line for aqueous sucrose and a gray dashed-dotted line for aqueous $NH_4NO_3$. It should be noted that the measurement range for aqueous binary mixtures is limited to higher water activities (> 0.3). To indicate the uncertainty of these predictions, the region below 0.3 has been shaded in light gray. We only show the results of one example of the mass-fraction-based mixing rule in the supplementary information (see Fig. A2), since the results are very similar to the ZSR approach.

To quantitatively compare the mixing rules and AIOMFAC-VISC, we calculated the mean absolute error (MAE) between the measured data and the mixing rules and AIOMFAC-VISC:

$$MAE = \frac{1}{N} \sum_{i=1}^{N} |\log_{10}(y_i) - \log_{10}(x_i)|, \tag{6}$$





**Figure 5. Predictions of ternary mixture log₁₀(viscosity/(Pa·s)) as a function of water activity of sucrose–$NH_4NO_3$–$H_2O$ with mixing rules and AIOMFAC-VISC at 293 K.** Panels **(A & B)** show the results of an OIR = 1 (blue) and panel **(C & D)** the results of OIR = 4 (pink). Panel **(A & C)** show the log₁₀(viscosity/(Pa·s)) calculated with the mole-fraction-based mixing (pink solid line) and the ZSR-based mixing approach (pink dotted line). Panel **(B & D)** show the predicted log₁₀(viscosities) of AIOMFAC-VISC for binary and ternary systems. The log₁₀(viscosities) obtained from the measurements (circles) are the same as those in Fig. 4. The left column shows the fitted curves for binary aqueous sucrose and aqueous $NH_4NO_3$ solutions by the gray-dashed and gray-dotted-dashed lines, respectively. The right column shows the predictions of AIOMFAC-VISC (a dashed gray line for aqueous sucrose, a dotted-dashed gray line for aqueous $NH_4NO_3$, and solid lines for mixed systems). Shaded areas indicate the AIOMFAC-VISC uncertainty. The area for $a_w \leq 0.3$ is shaded gray to indicate that the results of the mixing rule and the AIOMFAC-VISC model in this area are more uncertain, since no experimental data for aqueous $NH_4NO_3$ and aqueous sucrose are available in this range.



where $N$ is the number of data points, $y$ are the viscosity predictions of the mixing model and $x$ are the experimental mixture viscosity data. The results are shown in Table 2. We divided them into two regimes, above water activity of 0.3 and below water activity of 0.3 where binary aqueous viscosity data of $NH_4NO_3$ and sucrose are not available.

The molar mixing and AIOMFAC-VISC predictions for water activities greater than 0.3 show good agreement with the measured viscosities. The MAE for all scenarios is half an order of magnitude or less. However, the ZSR approach does not capture the data as well, with an MAE larger than one order of magnitude. For water activities below 0.3 and OIR = 4, molar mixing predictions have an MAE below one order of magnitude, while both ZSR and AIOMFAC-VISC exhibit several orders of magnitude deviations. For OIR = 1, a quantitative evaluation is not possible, as we only could determine the upper and lower limits of the viscosities. However, at a water activity of 0.1, AIOMFAC-VISC predicts viscosities several orders of magnitude higher than those indicated by our poke-flow measurements. The molar mixing rule may better predict this data point. However, with a water activity of 0.05, AIOMFAC-VISC better predicts the sharp increase compared to the mixing rule.

Since the mole-fraction-based mixing rule can reasonably well predict the viscosity of $NH_4NO_3$–sucrose–$H_2O$, we tested whether this also holds for mixtures of sucrose with other inorganic salts from prior studies. The results are shown in Figure 6. It can be seen that here also the viscosity prediction of the simple mole-fraction-based mixing of the aqueous binary systems is consistent with published data on ternary mixtures, at least for the water activities above 30% (MAE < 1). The predictions fits better to the viscosity data of monovalent ions than to those of divalent ions, where the predictions of the ZSR approach fit the best. Mixtures with divalent ions can undergo gel formation, as shown by Richards et al. (2020), which may be an explanation for the difference.

Overall, from the modest data set that we analyzed, we conclude that if the data of the aqueous binary systems are available, those can be used to predict the viscosity of the ternary mixtures in a simple and computationally very efficient manner. However, if binary data are not available, AIOMFAC-VISC can give predictions of the viscosity and that model extends naturally beyond ternary systems, as demonstrated for surrogate systems of secondary organic aerosols by Gervasi et al. (2020) and Lilek and Zuend (2022). The mole-fraction-weighted mixing rule may also apply to multi-component mixtures beyond ternary mixtures, but it has not yet been evaluated for such cases.





**Figure 6. $\log_{10}$(viscosity/(Pa·s)) as a function of water activity for ternary aqueous sucrose–inorganic mixtures at 293 K. (A & B)** show the viscosities of $NO_3$–sucrose–$H_2O$, **(C & D)** NaCl–sucrose–$H_2O$, **(E & F)** $(NH_4)_2SO_4$–sucrose–$H_2O$, **(G & H)** $Ca(NO_3)_2$–sucrose–$H_2O$ and $Mg(NO_3)_2$–sucrose–$H_2O$. Linear or quadratic fits of the aqueous inorganic salts are shown with black dashed dotted lines, and the fit of aqueous sucrose with a gray dashed line. The solid pink line is simple mole-fraction-based mixing, and the dotted pink line is ZSR-based mixing. In the gray area, the estimation of the mixing rule is uncertain due to the limited availability of binary data.





**Table 2.** Mean absolute error (MAE) for molar mixing, ZSR-based mixing and AIOMFAC-VISC for inorganic salts mixed with sucrose for water activities smaller and larger equal 0.3. Upper and lower limits of the poke-flow measurements (indicated with an upward or downward facing arrow) were not considered for the MAE.

| Inorganic salt | OIR | Molar Mixing $a_w \geq 0.3$ | ZSR $a_w \geq 0.3$ | AIOMFAC $a_w \geq 0.3$ | Molar Mixing $a_w < 0.3$ | ZSR $a_w < 0.3$ | AIOMFAC $a_w < 0.3$ |
|---|---|---|---|---|---|---|---|
| $NH_4NO_3$ | 1 | 0.17 | 1.33 | 0.37 | - | - | - |
| $NH_4NO_3$ | 4 | 0.22 | 0.92 | 0.52 | 0.76 | 3.81 | 4.76 |
| $NaNO_3$ | 4 | 0.71 | 0.70 | | - | - | |
| $NaNO_3$ | 1.5 | 0.29 | 1.10 | | 1.84 | 2.02 | |
| $(NH_4)_2SO_4$ | 4 | 0.32 | 0.44 | | 0.59 | 1.35 | |
| $(NH4_4)_2SO_4$ | 1 | 0.46 | 1.13 | | 5.64 | 3.60 | |
| $NaCl$ | 6 | 0.55 | 0.76 | | - | - | |
| $NaCl$ | 23 | 0.37 | 0.31 | | - | - | |
| $Ca(NO_3)_2$ | 1 | 0.48 | 0.24 | | 2.24 | 1.40 | |
| $Mg(NO_3)_2$ | 1 | 0.13 | 0.10 | | 2.63 | 3.44 | |
| **Total** | | **3.7** | **7.03** | | **13.66** | **15.62** | |

## 3.3 Atmospheric Implications

Our findings allow us to estimate the range of relative humidities for which $NH_4NO_3$ partitioning under typical atmospheric conditions is limited by condensed phase ion transport. With this goal, we estimate the internal mixing times (Eq. 2) for
$NH_4NO_3$-containing aqueous organic aerosol particles and compare these with the mixing times for aqueous binary organic particles. Global models often assume equilibrium partitioning is achieved for fine particulate matter within the typical model time steps used for periodic output (tens of minutes to ca. 1 hour, e.g. Bian et al. (2017)). If the mixing times exceed the chemical time step in global models, it makes the quasi-instantaneous equilibrium assumption questionable.

For our estimates, we use the following assumptions and simplifications. As organic compounds, we consider two cases: (i)
for a high–viscosity example sucrose as a single compound and (ii) for a more moderate example SOA derived from toluene oxidation, for which data of the viscosity are available (Song et al., 2016a). This SOA viscosity as a function of water activity was taken from (Maclean et al., 2021). The mixing times are calculated for a temperature of 293 K, which is a representative temperature of the summer months in the PBL. Since the characteristic mixing time scales with the square of the particle radius, regardless of the composition, we consider only a single size, with the radius of the particles set to 100 nm. The
diffusion coefficients are determined from the viscosities via the Stokes–Einstein relation (Eq. 1), assuming an unhydrated hydrodynamic radius for ammonium ions of 161 pm (Zuend et al., 2008). We assume that the ammonium ion primarily limits the diffusivity and not the nitrate ion, because the hydrodynamic radius of the latter is smaller (Zuend et al., 2008). The



Stokes–Einstein relation provides a lower limit for the estimated diffusion coefficient in cases for which the effective sizes of the diffusing molecules or ions are lower than the size of the matrix molecules (Evoy et al., 2020). Based on the O:C ratio of the

organic component, we assume that the ternary systems $NH_4NO_3$–sucrose–$H_2O$ (O:C = 0.92) and $NH_4NO_3$-toluene–derived SOA-$H_2O$ (O:C on average 1.1, Li et al. (2015)) are both not phase-separated. To determine the viscosities of the mixed systems, we used the mole-fraction-based mixing approach. To determine the dry OIR we used the compositions provided by Zhang et al. (2011) for Zurich summer (OIR = 2), Beijing (OIR = 0.53) and Mexico City (OIR = 1.7), assuming for simplicity that all inorganics are $NH_4NO_3$ and all organics are sucrose or toluene-derived SOA (average molar mass 350 $g\,mol^{-1}$, Hinks

et al. (2018)), respectively. Note, that the OIR's observed in Zurich summer are similar to rural/remote regions reported in the same study (Zhang et al., 2011).

The estimated internal mixing times are shown in Fig. 7. Panel 7A shows $NH_4NO_3$ mixed with sucrose, and panel 7B $NH_4NO_3$ mixed with toluene–derived SOA. In all cases, the mixing times of the inorganic–organic mixed systems, assumed to form a single condensed phase, are orders of magnitude lower than the mixing times of the aqueous organic particle. Most importantly,

the mixing times for inorganic–organic particles with radius of 100 nm stay orders of magnitude below one hour even for RH below 30%. Note, that the RH of the PBL is often above 25% (Maclean et al., 2017). Hence, our findings support the assumption that for these particles the equilibrium partitioning is fulfilled, at least for boundary layer temperatures. To be more specific, while the mixing times for aqueous organic particles exceed the mixing times of 1 hour at RH lower than approximately 35%, the corresponding time for particles containing inorganic compounds decreases to less than a second.

Zurich (summer) and Mexico City have similar compositions and therefore similar mixing times. Beijing has a smaller organic fraction, resulting in up to two orders of magnitude differences in mixing times compared to Zurich (summer) and Mexico City.

There are several factors that influence these mixing times, most importantly, the temperature and possibly also the morphology of the aerosol particles. For a discussion of the influence of morphology, in particular liquid–liquid phase separation, see Lilek

and Zuend (2022). Briefly, liquid–liquid phase separation can result in a thin but highly viscous, organic-rich outer shell phase for an aqueous particle containing inorganic ions. This leads to long characteristic mixing times, even if this shell is thin and the viscosity of the inner aqueous inorganic core low.

Since the temperature in the troposphere typically decreases with altitude and the viscosity increases with decreasing temperature, we expect the mixing times to increase at altitudes above the PBL. For a quantitative estimate of the effect of temperature

on the mixing times, we need to estimate the viscosities of the ternary mixtures. An attempt to parameterize the temperature dependence for the system shown in Fig. 7B is detailed in Appendix B. It is based on known temperature dependent viscosity data for water and subcooled $NH_4NO_3$ melt data as well as the prediction of temperature dependent viscosity of aqueous toluene-derived SOA by Gervasi et al. (2020). In addition to these estimates for the lower temperatures of the binary systems, we assume that the mole-fraction-based viscosity mixing rule that has been shown to yield reasonable data at elevated tem-

peratures also holds at lower temperatures. Calculated characteristic mixing times for 100 nm radius particles of the ternary



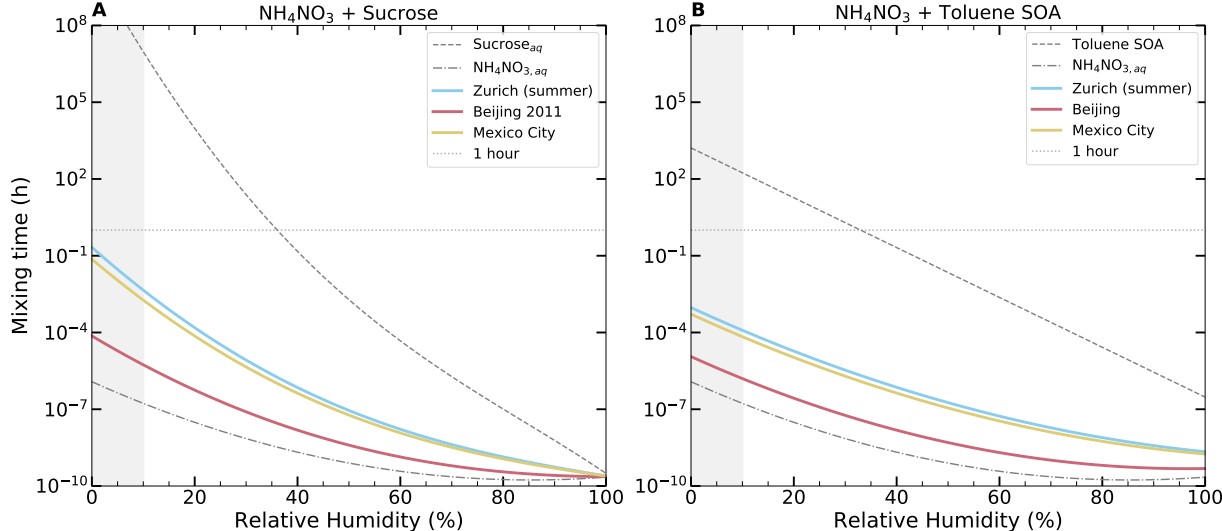

**Figure 7. Characteristic internal mixing times of** $NH_4^+$**-ions as a function of RH in 100 nm radius particles at 293 K consisting of the ternary system** $NH_4NO_3$**–organics–**$H_2O$**.** Panel (**A**) $NH_4NO_3$–sucrose–$H_2O$ and panel (**B**) $NH_4NO_3$–toluene–derived SOA– $H_2O$. The mixing times of the ternary inorganic–organic mixtures determined by mole-fraction-based mixing of the aqueous binary compounds are shown as colored solid lines. The mixing time of the aqueous $NH_4NO_3$ particle is shown as a gray dotted-dashed line. The mixing times of the aqueous sucrose and aqueous toluene–derived SOA (Maclean et al., 2021) particles are shown as gray-dashed lines. We consider the compositions of three cities as reported by Zhang et al. (2011) assuming that all organics are sucrose/toluene–derived SOA and all inorganics are $NH_4NO_3$: Zurich summer (cyan, OIR = 2), Beijing (rose, OIR = 0.53), and Mexico City (sand, OIR = 1.7). A one-hour characteristic internal mixing time is indicated with a horizontal gray dotted line. The gray-shaded area indicates uncertain results due to unavailable aqueous binary data.

$NH_4NO_3$–toluene-derived SOA system (OIR=2) remain below 1 hour over almost the entire tropospheric temperature range down to colder than 230 K for relative humidities larger than 30 %. This would indicate that the assumption of equilibrium partitioning for $NH_4NO_3$ in such an aerosol is a good approximation. We would like to emphasize, however, that these estimates are not yet supported by low-temperature measurements and have been derived by considerable extrapolation from
existing data. They should therefore be treated with an appreciation for increasing uncertainties toward lower temperatures accompanying these extrapolations and the potential need for revisions in future. In any case, this finding highlights the need to extend experimental studies like ours to lower temperatures to gain a better understanding of whether equilibrium holds for tropospheric inorganic–organic aerosol particles.

## 4 Conclusion

In this study, we investigated the viscosity of inorganic–organic ternary systems of $NH_4NO_3$, sucrose, and water over the full range of RH at room temperature for OIR of 1 and 4. The viscosity of sucrose mixed with $NH_4NO_3$ was shown to remain



liquid ($< 10^2$ Pa s) at an RH greater than 45% and is up to two orders of magnitude lower than previously reported (Tong et al., 2022)). Only under low humidity conditions (RH < 10%) the ternary mixture becomes glassy, in contrast to binary aqueous sucrose, which becomes glassy at a higher humidity (RH < 28%).

We conclude that if binary viscosity data of the aqueous organic solution and aqueous inorganic solution are available, a simple mole-fraction-based mixing rule of the logarithms of pure-component viscosities seems to be sufficient to predict the viscosity of the ternary system when the system remains in a homogeneous single-phase state. However, if these data are not available, AIOMFAC-VISC can provide a reasonable estimate. Measuring additional mixed inorganic–organic systems could allow us to obtain a more complete database to validate AIOMFAC-VISC, to thoroughly evaluate its different viscosity

mixing approaches, and to inform about the optimal approach for a range of compound classes and mixtures covering different electrolyte components.

Since the available experimental techniques are limited in their applicable viscosity range, particularly at viscosities above $10^8$ Pa s, aqueous phase viscosities of some organic compounds or related mixtures at low RH cannot be measured. However, by mixing organics with monovalent inorganic salts, the viscosities of those systems may become accessible at low RH. This may

offer a measurement strategy to obtain the much-needed data to further validate predictive models such as AIOMFAC-VISC.

With respect to characteristic internal mixing times for aerosol particles, we show that those of internally mixed single-phase inorganic–organic aerosol particles can be orders of magnitude lower than those of electrolyte-free aqueous SOA. Assuming instantaneous equilibrium partitioning is justifiable in the PBL for these particles if the temperature is near 293 K, even if the SOA on its own were highly viscous or glassy under the same conditions.

Interestingly, our estimates indicate that mixing times do not exceed one hour even for mid-latitude tropospheric temperatures if the relative humidity is larger than 40%. However, this finding is based on a number of assumptions and extrapolations and calls for additional viscosity measurements at lower temperatures. In addition, gas–particle partitioning experiments with $NH_4NO_3$, which can be performed at low temperatures, can be used to to derive diffusivities in organic mixtures and to directly test the mass transport limitations.

*Code availability.* An online version of AIOMFAC-VISC is available at http://www.aiomfac.caltech.edu/model.html. The source code of AIOMFAC-VISC is available on GitHub (last access: 2 May 2024) (https://github.com/andizuend/AIOMFAC, Zuend et al., 2021; https://doi.org/10.5281/zenodo.6049217, Zuend (2022)).

*Data availability.* Data will be made available upon acceptance of the manuscript



*Author Contribution.* L.K.K, A.K.B, U.K.K conceptualized this work. All co-authors contributed to writing this manuscript.
A.K.B, A.Z., F.G, and U.K.K. edited the manuscript. L.K.K. performed the experiments, and analyzed the data with contributions of F.G.. A.Z. performed the AIOMFAC-VISC modeling.

*Competing interests.* No competing interests are present.

# 5 Acknowledgements

This work was funded by the Swiss National Science Foundation (grant number 189939). A.K.B. acknowledges support from the Natural Sciences and Engineering Research Council of Canada (RGPIN-2023-05333). A.Z. acknowledges support by the Alfred P. Sloan Foundation (grant no. G-2020-13912) and the Natural Sciences and Engineering Research Council of Canada (grant no. RGPIN-2021-02688). The authors acknowledge the assistance of personnel from the UBC BioProducts Institute and the Canada Foundation for Innovation (Project number 38623) and the viscosity data provided by Prof. Mijung Song. We acknowledge Thomas Peter and Nir Bluvshtein for fruitful discussions and feedback on the manuscript. In addition, we acknowledge deepL.com and Writefull which were used to improve the language and grammar of this manuscript.



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



# Appendix A: Aqueous binary mixture fit coefficients

**Table A1.** Parameters of the fits for the viscosity of binary aqueous inorganic in the form of $\log_{10}(\eta/[\mathrm{Pa\,s}]) = a\,a_w{}^2 + b\,a_w + c$.

| Compound | a | b | c |
|---|---|---|---|
| $NH_4NO_3(aq)$ | 5.2571978 | -8.98631907 | 0.75186241 |
| $NaNO_3(aq)$ | 0 | -3.99991178 | 0.92733166 |
| $NH4_2SO_4(aq)$ | 0 | -3.89866574 | 0.92739022 |
| $NaCl(aq)$ | 0 | -2.23773231 | -0.76170413 |
| $CaNO_3(aq)$ | 3.80725639 | -9.57730725 | 2.85643819 |
| $MgNO_3(aq)$ | -0.67419009 | -0.9381783 | -1.33926278 |

For sucrose a quartic polynomial equation is fitted ($\log_{10}(\eta/[\mathrm{Pa\,s}]) = a\,a_w{}^4 + b\,a_w{}^3 + c\,a_w{}^2 + d\,a_w + e$ with a = -2.63720583, b = -5.66469305, c = 26.11320604, d = -37.8954176, e=17.25426411).

To estimate the viscosity of the subcooled $NH_4NO_3$ melt at atmospheric temperatures, we fit the melt viscosity data from Booth and Vinyard (1967) to the modified Vogel–Tamman–Fulcher (VTF) equation (Angell, 1991) and set the viscosity at the glass transition temperature ($T_g$ = 207.5 K; Angell and Helphrey (1971)) to $10^{12}$ Pa·s (Angell, 1991). This results in the following parameterization:

$$\log_{10}(\eta/[\mathrm{Pa\,s}]) = -5 + \frac{0.434 \cdot T_0 \cdot D}{T - T_0}, \quad T_0 = 39.17 \frac{T_g}{D + 39.17}, \tag{A1}$$

with the fragility parameter $D = 10.57$. This results in a viscosity of 6.0 Pa·s of the pure subcooled $NH_4NO_3$ melt at 293 K.

# Appendix B: Estimation of temperature dependent atmospheric mixing times for $NH_4NO_3$ - toluene–derived SOA mixtures

We combine the estimation for the temperature-dependent viscosity of the subcooled melt of $NH_4NO_3$ (Eq. A1) with an equivalent fit to the known temperature dependence of the viscosity for pure water (Dehaoui et al., 2015). Applying mol-fraction-based mixing to these two parameterizations yields temperature and water activity dependent viscosity predictions for aqueous $NH_4NO_3$, as shown by the solid lines in Fig. B1. A comparison of these predictions at 290 K with the data and fit of Fig. 3(a) shows that these predictions tend to overestimate the viscosity at intermediate water activities – at least at typical PBL temperatures.





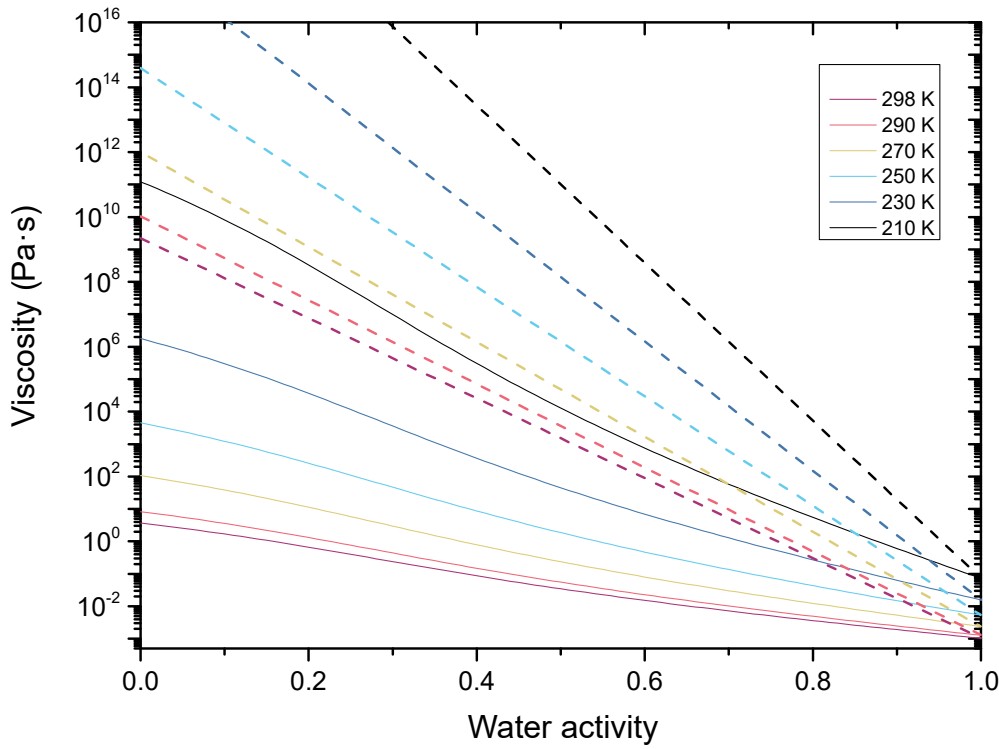

**Figure B1. Viscosity estimates as function of water activity for binary, aqueous** $NH_4NO_3$ **(solid lines) and aqueous toluene–derived SOA (dashed lines), color-coded for temperatures ranging from 298 K to 230 K.**

For aqueous toluene–derived SOA, we again fit a VTF type equation to the available data (Song et al., 2016b) and the AIOMFAC-VISC predictions (Gervasi et al., 2020). These fits (shown in Fig. B1 as dashed lines) are in close agreement (within 1 order of magnitude) with both the data of Song et al. (2016b) and the temperature-dependent predictions by Gervasi
et al. (2020), as well as the fit given in Maclean et al. (2021).




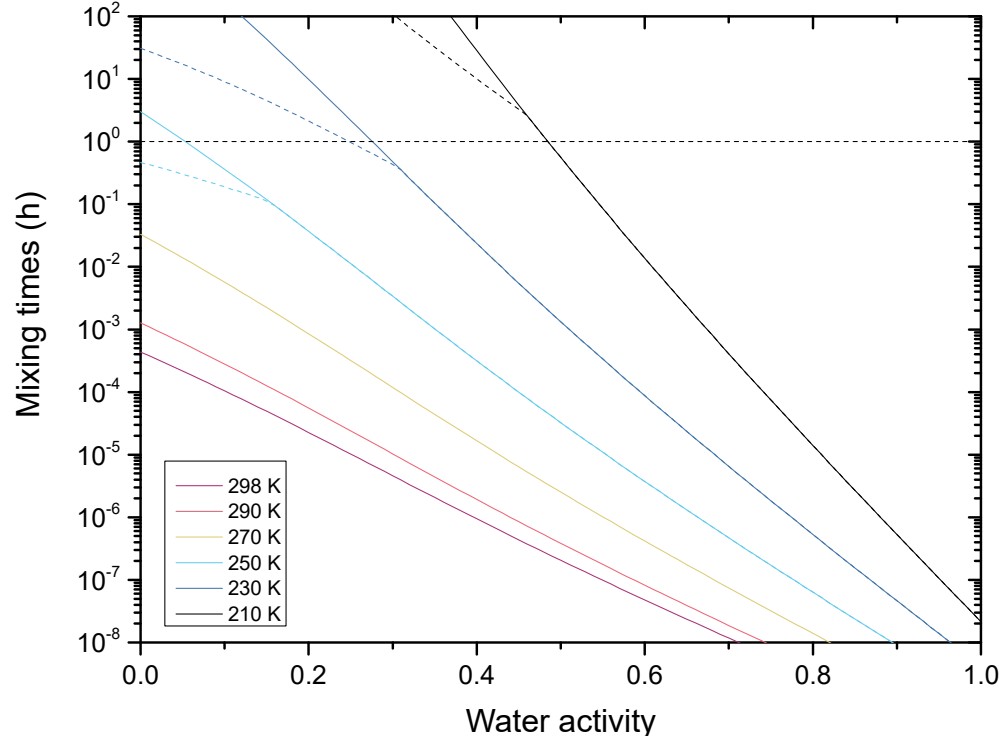

**Figure B2. Mixing times ($\tau$) as function of water activity for aqueous toluene–derived SOA mixed with aqueous** $NH_4NO_3$ **for a dry mass ratio of 2:1 and temperatures ranging from 298 K to 210 K.** These times were calculated for a 100 nm radius particle assuming an unhydrated hydrodynamic radius for the ammonium ion of 1.61 Å. The dashed horizontal line marks one hour mixing time. The dashed lines for 250 K, 230 K and 210 K are calculated assuming that the viscosity of the binary systems cannot exceed $10^{12}$ Pa·s.

In order to calculate the characteristic mixing times for toluene–derived SOA mixed with $NH_4NO_3$, we take the viscosities for the binary aqueous systems of Fig. B1 and assume for the entire temperature range that the mole-fraction-based mixing rule for the viscosity of a pseudo-ternary mixture (approximating this multi–component system) holds based on the binary subsystems. With these assumptions, we predict the characteristic mixing times shown in Fig. B2. These predicted mixing times are faster than 1 hour for almost the entire temperature range of the troposphere if the relative humidity exceeds 40%. It should be noted, however, that the predicted mixing times are based on numerous assumptions and extrapolations as outlined, and need to be supported by experiments before drawing firm conclusions.




## Appendix C: Calculation of equilibration times for the laboratory experiments & tabulated results

To estimate the equilibration time of the water in the droplets to reach equilibrium with the gas phase RH in the poke-flow
and rFRAP experiments, we need to estimate the diffusivity of water in the $NH_4NO_3$–sucrose–$H_2O$ mixtures. Assuming a
Vinges-type equation (Vignes, 1966), we can estimate the diffusivity of water in the ternary mixture $D_{H_2O,mix}$ based on the
diffusivities in the two binary solutions:

$$D_{H_2O,mix}(a_w) = D_{H_2O,S_{aq}}(a_w)^{(1-x)} D_{H_2O,NH_4NO_{3aq}}(a_w)^x \qquad (C1)$$

where $D_{H_2O,S_{aq}}$ is the diffusivity of water in aqueous sucrose, $D_{H_2O,NH_4NO_{3aq}}$ is the diffusivity of water in aqueous $NH_4NO_3$,
and $x$ is the dry mole fraction of $NH_4NO_3$. $D_{H_2O,S_{aq}}$ at water activities below 0.1 is approximately $10^{-17}$ m$^2$ s$^{-1}$ (Zobrist
et al., 2011). $D_{H_2O,NH_4NO_{3aq}}(a_w{=}0.1)$ is approximately $3.7{\cdot}10^{-11}$ m$^2$ s$^{-1}$ and $D_{H_2O,NH_4NO_{3aq}}(a_w{=}0.2)$ is approximately
7.3 $\cdot10^{-11}$ m$^2$ s$^{-1}$ using the prediction of the viscosity for aqueous $NH_4NO_3$ by AIOMFAC-VISC, which can be related
to the diffusivity using the Stokes-Einstein relation ($\eta(a_w = 0.1) \approx 0.04$ Pa s, $\eta(a_w = 0.2) \approx 0.02$ Pa s, $r = 1.4e - 10$ m, and
$T = 293.15$ K). The dry mass fractions of 1 and 4 converted into mole fractions are $x = 0.81$ and $0.52$, respectively. Substituting
these values into Equation C1, the diffusivities of water in the $NH_4NO_3$–sucrose–$H_2O$ mixture can be calculated (see results
in Table C1). With these diffusion coefficients, we can estimate the required equilibration times $\tau_{calc}$ using equation 2 (refer
to Table C1). To verify the accuracy of our estimated equilibration times experimentally, we performed poke-flow experiments
after two different sample equilibration times $\tau_{eq}$ at 20% RH ($\tau_{eq,1} = 2$ h and $\tau_{eq,2} = 6$ h) and 16% RH ($\tau_{eq,1} = 18$ h and $\tau_{eq,2}$
= 96 h). The results of these experiments were similar, indicating that the equilibration times are satisfactory (see Table C1).

**Table C1.** Experimental and calculated equilibration times for the poke-flow experiments at all tested conditions, as well as results for the
obtained viscosities at 293 K.

| OIR | RH (%) | $\tau_{eq}$ (h) | $\tau_{calc}$ (h) | $D_{H_2O, mix}$ (m$^2$ s$^{-1}$) | $d_{max}$ (μm) | $\log_{10}(\eta(\text{Pa s}))$ | N |
|-----|--------|-----------------|-------------------|----------------------------------|----------------|--------------------------------|---|
| 1 | 10 | > 4 | < 1 | 2.4E-12 | 200 | $\leq 3$ | 5 |
| 1 | 5 | > 4 | < 1 | 1.6E-12 | 200 | $\geq 8.5$ | 3 |
| 4 | 23 | 14 | 5 | 3.2E-14 | 149 | $2.69 \pm 0.58$ | 3 |
| 4 | 20 | 2 | 4 | 3.2E-14 | 128 | $4.37 \pm 0.33$ | 3 |
| 4 | 20 | 6 | 3 | 3.2E-14 | 114 | $4.41 \pm 0.71$ | 3 |
| 4 | 16 | 18 | 9 | 2.2E-14 | 167 | $6.16 \pm 0.53$ | 4 |
| 4 | 16 | 96 | 8 | 2.2E-14 | 163 | $6.16 \pm 0.33$ | 3 |
| 4 | 10 | 22 | 5 | 2.2E-14 | 127 | $7.43 \pm 0.49$ | 3 |
| 4 | 5 | 24 | 7 | 1.7E-14 | 127 | $> 8.5$ | 4 |




**Table C2.** rFRAP experimental parameters and calculated diffusion coefficients of Rhodamine B isothiocyanate dextran in $NH_4NO_3$–sucrose—$H_2O$ and the corresponding viscosities at 293 K.

| OIR | RH (%) | $\tau_{eq}$ (h) | $\tau_{calc}$ (h) | $D_{dye}$ (m²/s) | $\log_{10}(\eta_{mean}(Pa\,s))$ | $\log_{10}(\eta_{max}(Pa\,s))$ | $\log_{10}(\eta_{min}(Pa\,s))$ | N | $d_{max}$ (µm) |
|---|---|---|---|---|---|---|---|---|---|
| 1 | 43 | 71 | < 1 | 2.18E-13 | -0.74 | -0.53 | -0.87 | 6 | 1000 |
| 1 | 59 | 22 | < 1 | 1.36E-12 | -1.57 | -1.54 | -1.59 | 5 | 1000 |
| 4 | 43 | 72 | < 20 | 4.66E-16 | 2.11 | 2.30 | 1.57 | 3 | 1000 |
| 4 | 59 | 22 | < 20 | 4.46E-14 | -0.01 | 0.15 | -0.35 | 4 | 1000 |
| 4 | 75 | 47 & 72 | < 20 | 1.19E-12 | -1.37 | -1.16 | -1.84 | 8 | 1000 |

**Table C3.** Experimental parameters and results for the viscometry measurements.

| OIR | RH (%) | T (K) | mean($\eta$ (Pa s)) | sd($\eta$ (Pa s)) | N | Measurement Device |
|---|---|---|---|---|---|---|
| 1 | 63.5± 2 | 298 | 1.65E-02 | 2.61E-04 | 3 | Rheometer (MCR302, Anton-Parr) |
| 1 | 67.8± 0 | 293 | 1.24E-02 | 1.73E-04 | 3 | Viscometer (LVDV-II, Brookfield) |
| 1 | 73.7± 0 | 293 | 7.72E-03 | 2.57E-04 | 3 | Viscometer (LVDV-II, Brookfield) |
| 4 | 75.6± 2 | 298 | 5.03E-02 | 1.41E-04 | 3 | Rheometer (MCR302, Anton-Parr) |
| 4 | 86.4± 2 | 298 | 1.02E-02 | 2.98E-05 | 3 | Rheometer (MCR302, Anton-Parr) |
| 4 | 93.0± 2 | 298 | 2.82E-03 | 4.14E-04 | 3 | Rheometer (MCR302, Anton-Parr) |





**Figure A1.** Microscopy image taken with a laser scanning microscope (z-stack) to determine the contact angle of a sucrose – $NH_4NO_3$ –$H_2O$ (OIR = 4) solution droplets under room conditions (Rhodamine 6G as dye). The middle shows the top view of the droplet. The right and upper panels show the cross-section.





**Figure A2. Predictions of ternary mixture log$_{10}$(viscosity/(Pa·s)) as a function of water activity of sucrose–$NH_4NO_3$–$H_2O$ with the ZSR approach and mass-fraction-based mixing at 293 K.** The log$_{10}$(viscosity/(Pa·s)) (OIR = 4) calculated with the ZSR approach and the mass-fraction-based mixing are shown with a pink dotted line and purple dash dotted line. The log(viscosities) obtained from the measurements (circles) are the same as those in Fig. 4. The fitted curves for the log$_{10}$(viscosity/(Pa·s)) of the binary aqueous sucrose and aqueous $NH_4NO_3$ solutions are gray-dashed and gray-dotted-dashed lines, respectively. The area of $a_w \leq 0.3$ is shaded gray to indicate that the results of the mixing rule and the AIOMFAC-VISC model in this area are more uncertain, since no experimental data for aqueous $NH_4NO_3$ and aqueous sucrose are available in this range.