# Peer review of "Viscosity of aqueous ammonium nitrate–organic particles: Equilibrium partitioning may be a reasonable assumption for most tropospheric conditions"

_EGUsphere, 2024_

## Referee Comment (RC2)

**Review for "Viscosity of aqueous ammonium nitrate–organic particles: Equilibrium partitioning may be a reasonable assumption for most tropospheric conditions" by Klein et al.**

Klein et al. measured the viscosity of particles with internally mixed $NH_4NO_3$ and sucrose using three techniques at atmospherically relevant humidity. The authors then predicted the viscosity based on mixing rules and the AIOMFAC-VISC model. They found that the mixing rule based on mole fractions is sufficient to predict the viscosity of a ternary system, e.g., the internally mixed $NH_4NO_3$, sucrose and $H_2O$. As viscosity is a very important property of aerosols and the observation data showing how the viscosity changes with particle composition (e.g, mixed inorganics and organics) are still very limited, this study is helpful in understanding the viscosity of internally mixed particles. However, I have some concerns about the authors' understanding of gas-particle partitioning. I recommend the publication of this study after the following comments could be addressed.

**Major comments:**

(1) My major comment arises from that the authors may have mixed the definition of mixing time and equilibrium time. Mixing time and equilibrium time are two different time scales. The equilibrium time refers to the time scale to achieve gas-particle equilibrium, affected by many factors, including particle viscosity, the volatility of partitioning compounds, particle size and concentrations etc, as gas-particle partitioning is controlled by gas-phase diffusion, interfacial transport, and particle-phase diffusion (Mai et al., 2015; Shiraiwa and Seinfeld, 2012; Li and Shiraiwa, 2019). However, the mixing timescale is mainly related to the particle-phase diffusion, as the authors wrote in Equation (2) in this study. Therefore, I suggest the authors be careful when using the term of "equilibrium time", e.g., the section of Appendix C, where the equilibration times are actually mixing timescales (Line 603, Line 616-619, Table C1).

(2) The authors applied three mixing rules, i.e., the mole-fraction based, the mass fraction based, and ZSR, to predict viscosity of the $NH_4NO_3$-sucrose-$H_2O$ system. Many other studies also applied the Gordon-Taylor equation combined with the VTF equation to predict the viscosity of particles with mixed compounds (Dette and Koop, 2015; Li et al., 2020; O'brien et al., 2021; Koop et al., 2011). As the values of $T_g$ of $NH_4NO_3$, sucrose, and $H_2O$ are available, it is possible to estimate the viscosity based on the Gordon-Taylor equation. Would the predictions using the Gordon-Taylor equation agree with the predictions based on the mole-fraction based mixing rule?

(3) In the section of 3.3 for Atmospheric Implications, the authors stated that "global models often assume equilibrium partitioning is achieved for fine particulate matter… if the mixing times exceed the chemical time step, it makes the quasi-instantaneous equilibrium assumption questionable". I do not agree with

this statement. Note that current chemical transport models (CTMs) often assume SOA partitioning is rapid, i.e., instantaneous equilibrium partitioning (Pankow, 1994) is usually employed for the gas-particle partitioning of semi-volatile organic compounds (SVOCs) forming SOA particles. When particle is viscous, however, the equilibrium timescale of SVOCs can be longer than 1 h, in which case kinetic partitioning of SVOCs should be considered instead of instantaneous equilibrium partitioning (Li and Shiraiwa, 2019; Maclean et al., 2021; Zhang et al., 2024). For $NH_4NO_3$ partitioning, however, I think there is no argument questioning the assumption of instantaneous equilibrium of $NH_4NO_3$ partitioning currently applied in CTMs. Therefore, I am wondering the meaningfulness of calculating the mixing timescales of $NH_4NO_3$ partitioning in fine particles. In the abstract, the authors should clearly state that they actually calculated the mixing time of $NH_4NO_3$ in internally mixed particles and in the paper title, the authors should clarify that equilibrium partitioning may be a reasonable assumption for $NH_4NO_3$ partitioning.

In addition, at Line 314, the authors wrote "the mixing time scales with the square of the particle radius, regardless of the composition". This is not correct as the mixing timescales do relate to the composition because the bulk diffusion coefficients are different for different diffusing compounds.

**Specific comments:**

(1) Line 90: I could not find the SI file attached?

(2) In the caption of Figure 3, is Tong et al. 2018 actually Tong et al. 2022?

(3) Line 251-253, the authors concluded that the viscosity values reported in Tong et al. (2022) for $NH_4NO_3$- $H_2O$ (Fig. 4) are questionable due to the volatilization of $NH_4NO_3$. Does this mean their data for the pure $NH_4NO_3$ showed in Fig. 3A are also questionable? Their data look comparable to Laliberté(2007) in Fig. 3A.

(4) Line 258, in your system, what is the possible reason for the significant increase in viscosity below $a_w$ of 0.1?

(5) Line 315-316, the authors calculated the diffusion coefficients of $NH_4NO_3$ via the Stokes-Einstein relation. However, previous studies have showed that the Stokes-Einstein relation is not suitable (with large underestimation) to predict the diffusion coefficients for such small diffusing molecules (Evoy et al., 2020; Price et al., 2015). Please also double check the calculation of $D_{H2O}$ (Line 613) in Equation (C1).

(6) Line 200: "$10^{12}$ Pa" should be "$10^{12}$ Pa s".

(7) Caption of Figure 6: "the viscosities of $NO_3$–sucrose–$H_2O$" should be "$NaNO_3$–sucrose–$H_2O$".

(8) Table A1, what temperature the viscosity parameterization showed in Table A1 is suitable to?

(9) The mixing times showed in Fig. B2 are for which compound? The mixing times of different molecules are very different (refer to my major comment 3).

(10) Table C2, what $d_{max}$ is? Is it the diameter of film?

**References**

1.   Dette, H. P. and Koop, T.: Glass Formation Processes in Mixed Inorganic/Organic Aerosol Particles, The Journal of Physical Chemistry A, 119, 4552-4561, 10.1021/jp5106967, 2015.

2.   Evoy, E., Kamal, S., Patey, G. N., Martin, S. T., and Bertram, A. K.: Unified Description of Diffusion Coefficients from Small to Large Molecules in Organic–Water Mixtures, The Journal of Physical Chemistry A, 124, 2301-2308, 10.1021/acs.jpca.9b11271, 2020.

3.   Koop, T., Bookhold, J., Shiraiwa, M., and Poschl, U.: Glass transition and phase state of organic compounds: dependency on molecular properties and implications for secondary organic aerosols in the atmosphere, Phys. Chem. Chem. Phys., 13, 19238-19255, 10.1039/C1CP22617G, 2011.

4.   Li, Y. and Shiraiwa, M.: Timescales of secondary organic aerosols to reach equilibrium at various temperatures and relative humidities, Atmos. Chem. Phys., 19, 5959-5971, 10.5194/acp-19-5959-2019, 2019.

5.   Li, Y., Day, D. A., Stark, H., Jimenez, J. L., and Shiraiwa, M.: Predictions of the glass transition temperature and viscosity of organic aerosols from volatility distributions, Atmos. Chem. Phys., 20, 8103-8122, 10.5194/acp-20-8103-2020, 2020.

6.   Maclean, A. M., Li, Y., Crescenzo, G. V., Smith, N. R., Karydis, V. A., Tsimpidi, A. P., Butenhoff, C. L., Faiola, C. L., Lelieveld, J., Nizkorodov, S. A., Shiraiwa, M., and Bertram, A. K.: Global Distribution of the Phase State and Mixing Times within Secondary Organic Aerosol Particles in the Troposphere Based on Room-Temperature Viscosity Measurements, ACS Earth and Space Chemistry, 5, 3458-3473, 10.1021/acsearthspacechem.1c00296, 2021.

7.   Mai, H., Shiraiwa, M., Flagan, R. C., and Seinfeld, J. H.: Under what conditions can equilibrium gas–particle partitioning be expected to hold in the atmosphere?, Environ. Sci. Technol., 49, 11485-11491, 10.1021/acs.est.5b02587, 2015.

8.   O'Brien, R. E., Li, Y., Kiland, K. J., Katz, E. F., Or, V. W., Legaard, E., Walhout, E. Q., Thrasher, C., Grassian, V. H., DeCarlo, P. F., Bertram, A. K., and Shiraiwa, M.: Emerging investigator series: chemical and physical properties of organic mixtures on indoor surfaces during HOMEChem, Environmental Science: Processes & Impacts, 23, 559-568, 10.1039/D1EM00060H, 2021.

9.   Pankow, J. F.: An absorption model of gas-particle partitioning of organic-compounds in the atmosphere, Atmos. Environ., 28, 189-193, 10.1016/1352-2310(94)90094-9, 1994.

10.   Price, H. C., Mattsson, J., Zhang, Y., Bertram, A. K., Davies, J. F., Grayson, J. W., Martin, S. T., O'Sullivan, D., Reid, J. P., and Rickards, A. M.: Water diffusion in atmospherically relevant α-pinene secondary organic material, Chemical Science, 6, 4876-4883, 2015.

11.   Shiraiwa, M. and Seinfeld, J. H.: Equilibration timescale of atmospheric secondary organic aerosol partitioning, Geophys. Res. Lett., 39, 10.1029/2012GL054008, 2012.

12.   Zhang, Z., Li, Y., Ran, H., An, J., Qu, Y., Zhou, W., Xu, W., Hu, W., Xie, H., Wang, Z., Sun, Y., and Shiraiwa, M.: Simulated phase state and viscosity of secondary organic aerosols over China, Atmos. Chem. Phys., 24, 4809-4826, 10.5194/acp-24-4809-2024, 2024.

---

## Author Comment (AC1)

We thank reviewer #1 for providing a positive review and the suggestions to improve the manuscript. We answer the reviewers' questions and comments below (our reply in blue, reviewer's comments in black).

Liviana et al. used three experimental measurement techniques and various theoretical calculation methods to quantify the viscosity of the ammonium nitrate-sucrose-water ternary aerosol systems and estimate the characteristic internal mixing times of such systems. Overall, the study data is comprehensive, the research methods are abundant, the discussions are detailed, and the scientific reliability is high. However, some discussion can be added:

1. Lines 249-253: The discussion in the manuscript attributed the differences in viscosity measurement to the volatilization of NH4NO3. While this is a possible reason, the volatilization of ammonium nitrate at room temperature causing changes in the OIR is unlikely to generate such a significant viscosity error (the OIR may increase from 1:1 to close to 2:1, but is unlikely to increase to 4:1 as in the experimental conditions in Figure 4). It is suggested to add a discussion on systematic errors between optical tweezers and the measurement methods in the manuscript, as well as the influences of suspension droplets and bulk phase solutions on viscosity measurements in this part of the discussion.

As we point out in the manuscript (line 251), the most reliable and best-established viscosity measurements are those of bulk viscometry (which are limited towards large viscosity). As shown in Fig. 4 and pointed out in the text these clearly deviate from the tweezer's data of Tong et al. (2022). Therefore, we conclude that the data of Tong et al. (2022) most likely are erroneous. As we are no experts to judge possible systematic errors of optical tweezer viscosity measurements, we refrain from discussing possible errors of the experiment of Tong et al. (2022) in detail. We agree with the reviewer that a significant volatilization is needed to explain the data of Tong et al. and such a volatilization should have been noticed by Tong et al. as they are tracking radius of the particles in their experiments. Alternatively, the actual OIR was not the reported 1:1 ratio. We therefore will delete the last sentence of the paragraph (lines 252- 253) and just end the paragraph with the statement that the Tong et al. data are questionable.

2. Line 365-371: Based on the data presented in the manuscript, the method of estimating the viscosity of mixed particles using a mole-fraction-based mixing rule is indeed more reliable. Furthermore, it is expected that the authors will add a discussion on the following topics in the conclusion section: The current measurement and estimation methods for aerosol viscosity are actually showing quite large uncertainty (especially the poke-flow method), as shown in Fig. 3 and Fig. 5, where the differences under the same water activity conditions can reach two orders of magnitude. So, does the authors have any recommended measurement and prediction methods in the conclusion section? Or discuss the tolerance levels for quantifying aerosol viscosity?

We thank the reviewer for this comment. We tried to discuss these points in the paragraph starting on line 373 in the conclusion section. We will rephrase this paragraph following the reviewers' suggestions as follows; it will read:

"The available experimental techniques are limited in their applicable viscosity range, particularly at viscosities above $10^8$ Pa s and carry uncertainties of orders of magnitude in this range (see Fig. 4). Hence, aqueous phase viscosities of some organic compounds or related mixtures at low RH cannot be measured precisely. Our results here suggest that a strategy to obtain the much-needed data to further validate predictive models such as AIOMFAC-VISC at low water activities is to mix organics with monovalent inorganic salts. The viscosities of these mixed systems at low RH will carry sufficiently low uncertainties with the techniques presently available (cp. Fig 4) to constrain mixing rules. Such measurement data may further the development of predictive models for viscosity of complex, atmospherically relevant aerosol."

3. Line 14-17: The authors emphasized that throughout the mid-latitude troposphere, the viscosity of inorganic-organic mixed aerosols is relatively low, and the kinetic limitations of gas-particle partitioning can be ignored. However, the work of this study seems insufficient to support this point of view: First, as the authors have mentioned, if the particle is in the form of an organic coating, the timescale of gas-particle partitioning on a high-viscosity coating is likely to be considered, and such organic-coating particles may be ubiquitous in the troposphere; Secondly, the quantification work of this study is mainly based on nitrate particles, and it can be known from previous viscosity measurement data that nitrate could significantly reduce the viscosity of the mixing system, while other major inorganic salts of aerosols, such as sulfates, may not have such large reducing effect on viscosity (it is recommended that the authors compare the viscosity of nitrate and sulfate aqueous solutions at different water activities). In a word, the results of this study may not represent the actual atmospheric aerosols of various inorganic salts and organic compounds mixtures. It is suggested to soften the statement in the abstract section, which may be too categorical.

Thank you for these comments. Regarding the effect of organic coatings: the statements in the abstract refer to internally mixed single-phase particles (containing aqueous inorganic salts) only. We refrain from making a broader statement about the case of phase-separated particles with a viscous organic coating, which as the referee suggests may show comparably slower equilibration with the gas phase.

Regarding the difference between sulfate and nitrate: consider Fig. 6 of the paper and compare panels (E and F) for ammonium sulfate mixed with sucrose for OIR= 1 and OIR=4 respectively with the corresponding ones for ammonium nitrate at for example aw=0.4, the sulfate systems show a viscosity of about one order of magnitude larger compared to the nitrate ones. While this is a significant increase, mixing times will increase by the same factor

which is not sufficient to push them to a range in which the mixing times approach several minutes.

Nevertheless, we will take the advice of the reviewer and soften the last sentence in the abstract section as follows:

"Further data are needed to see whether this assumption may even hold for the entire troposphere at mid-latitudes and RH > 35 %."

---

## Author Comment (AC2)

We thank reviewer #2 for their comments, spotting several minor errors in the manuscript and the suggestions to improve the manuscript. We answer the reviewer's questions and comments below (our reply in blue, reviewer's comments in black).

Klein et al. measured the viscosity of particles with internally mixed $NH_4NO_3$ and sucrose using three techniques at atmospherically relevant humidity. The authors then predicted the viscosity based on mixing rules and the AIOMFAC-VISC model. They found that the mixing rule based on mole fractions is sufficient to predict the viscosity of a ternary system, e.g., the internally mixed $NH_4NO_3$, sucrose and $H_2O$. As viscosity is a very important property of aerosols and the observation data showing how the viscosity changes with particle composition (e.g, mixed inorganics and organics) are still very limited, this study is helpful in understanding the viscosity of internally mixed particles. However, I have some concerns about the authors' understanding of gas-particle partitioning. I recommend the publication of this study after the following comments could be addressed.

**Major comments:**
(1) My major comment arises from that the authors may have mixed the definition of mixing time and equilibrium time. Mixing time and equilibrium time are two different time scales. The equilibrium time refers to the time scale to achieve gas-particle equilibrium, affected by many factors, including particle viscosity, the volatility of partitioning compounds, particle size and concentrations etc, as gas-particle partitioning is controlled by gas-phase diffusion, interfacial transport, and particle-phase diffusion (Mai et al., 2015; Shiraiwa and Seinfeld, 2012; Li and Shiraiwa, 2019). However, the mixing timescale is mainly related to the particle-phase diffusion, as the authors wrote in Equation (2) in this study. Therefore, I suggest the authors be careful when using the term of "equilibrium time", e.g., the section of Appendix C, where the equilibration times are actually mixing timescales (Line 603, Line 616-619, Table C1).

We absolutely agree with the reviewer on the difference between equilibrium time and mixing time. We will carefully check and correct the revised manuscript in cases in which we had used equilibration time although we meant mixing time.

To clarify the wording in Appendix C: we add this sentence after line 605: "For these large particles mixing timescales should be equal to the equilibrium times scales due to the fast diffusion rate of water in the gas-phase and the high vapor pressure of water

(2) The authors applied three mixing rules, i.e., the mole-fraction based, the mass fraction based, and ZSR, to predict viscosity of the $NH_4NO_3$-sucrose-$H_2O$ system. Many other studies also applied the Gordon-Taylor equation combined with the VTF equation to predict the viscosity of particles with mixed compounds (Dette and Koop, 2015; Li et al., 2020; O'brien et al., 2021; Koop et al., 2011). As the values of $T_g$ of $NH_4NO_3$, sucrose, and $H_2O$ are available, it is possible to estimate the viscosity based on the Gordon-Taylor equation. Would the predictions using the Gordon-Taylor equation agree with the predictions based on the mole-fraction based mixing rule?

This is an interesting suggestion. However, the problem with the VTF equation to estimate viscosity using Gordon-Taylor to estimate the glass transition of a mixture is that you need not just the glass transition temperatures for the pure compounds, but also need to estimate the Gordon-Taylor constant and for the VTF equation the fragility parameter. If you take those suggested by Li et al. (2020), namely a Gordon-Taylor constant of $k_{GT}=2.5$ and a fragility parameter of $d=10$, the prediction for an ammonium nitrate/sucrose mixture with OIR=4 overestimates the measured viscosity by about 3 orders of magnitude at an intermediate water activity of 0.5. It needs a lower fragility parameter of about 7 and a significantly larger Gordon-Taylor constant of about 6 to come close to the experimental data. See Fig. below:

[Figure]

(3) In the section of 3.3 for Atmospheric Implications, the authors stated that "global models often assume equilibrium partitioning is achieved for fine particulate matter… if the mixing times exceed the chemical time step, it makes the quasi-instantaneous equilibrium assumption questionable". I do not agree with this statement. Note that current chemical transport models (CTMs) often assume SOA partitioning is rapid, i.e., instantaneous equilibrium partitioning (Pankow, 1994) is usually employed for the gas-particle partitioning of semi-volatile organic compounds (SVOCs) forming SOA particles. When particle is viscous, however, the equilibrium timescale of SVOCs can be longer than 1 h, in which case kinetic partitioning of SVOCs should be considered instead of instantaneous equilibrium partitioning (Li and Shiraiwa, 2019; Maclean et al., 2021; Zhang et al., 2024). For $NH_4NO_3$ partitioning, however, I think there is no argument questioning the assumption of instantaneous equilibrium of $NH_4NO_3$ partitioning currently applied in CTMs. Therefore, I am wondering the meaningfulness of calculating the mixing timescales of $NH_4NO_3$ partitioning in fine particles. In the abstract, the authors

should clearly state that they actually calculated the mixing time of $NH_4NO_3$ in internally mixed particles and in the paper title, the authors should clarify that equilibrium partitioning may be a reasonable assumption for $NH_4NO_3$ partitioning. In addition, at Line 314, the authors wrote "the mixing time scales with the square of the particle radius, regardless of the composition". This is not correct as the mixing timescales do relate to the composition because the bulk diffusion coefficients are different for different diffusing compounds.

We absolutely agree with the reviewer that when the particle is viscous the equilibrium time scale of SVOCs can be longer than 1 h and instantaneous equilibrium partitioning should no longer be used. That is what we intended to say when writing in line 306 ff: "Global models often assume equilibrium partitioning is achieved for fine particulate matter within the typical model time steps used for periodic output (tens of minutes to ca. 1 hour, e.g. Bian et al. (2017)). If the mixing times exceed the chemical time step in global models, it makes the quasi-instantaneous equilibrium assumption questionable." To make this clearer we will add "condensed phase" before "mixing times" in the last sentence.

We disagree with the reviewer that there is no argument questioning the assumption of instantaneous equilibrium of NH4NO3 partitioning. Imagine the following scenario: organic particles are in a polluted environment within the boundary layer and have sufficient time to reach equilibrium with ammonia and nitric acid at moderate temperatures. Now these particles experience a convective event with rapid updraft into the free troposphere where the gas phase concentrations of ammonia and nitric acid are low. If these organic particles containing ammonium nitrate would be sufficiently viscous (becoming more and more viscous with the lower temperatures of the updraft), the partitioning of ammonia and nitric acid back to the gas phase could potentially become kinetically limited. This is very similar to the scenario discussed in Bastelberger et al. (2017; Atmos. Chem. Phys., 17, 8453–8471), there for a semi-volatile organic instead of the equally semi-volatile ammonium nitrate. In the present manuscript we argue that this scenario indeed (as the reviewer writes) will not be likely to happen for ammonium nitrate as the viscosity of such mixed particles is quite small according to our measurements. But to our understanding this is not yet well established as measurements of the viscosity of inorganic/organic mixtures are scarce.

Referring to the question of the reviewer about the title: we indeed think that equilibrium partitioning is valid not only for ammonia and nitric acid but also for SVOCs as we show the addition of ammonium nitrate drastically reduces the viscosity of these particles as long as they are well mixed and do not exhibit phase separation.

We add the following two sentences at line 344 to make this clearer to the reader: "While the estimations for the mixing times are done for the ammonium ion, the same conclusion holds true for semi-volatile organic compounds (SOVCs) as long as the particles are single-phase and well mixed. They do not apply to the case of two-phase particles with an organic-rich phase."

We thank the reviewer for spotting that line 314 of the manuscript is misleading and apologize. We want to point out that for Fig. 7 we consider a single size only as mixing times scale with $r^2$ for a **fixed** composition. We will correct this incorrect phrasing in the revised manuscript.

**Specific comments:**

(1) Line 90: I could not find the SI file attached?

Thank you, we decided not to have a SI after all; in line 90 we should have referred to Appendix C. Corrected for the revised version.

(2) In the caption of Figure 3, is Tong et al. 2018 actually Tong et al. 2022?

Corrected for the revised version.

(3) Line 251-253, the authors concluded that the viscosity values reported in Tong et al. (2022) for $NH_4NO_3$- $H_2O$ (Fig. 4) are questionable due to the volatilization of $NH_4NO_3$. Does this mean their data for the pure $NH_4NO_3$ showed in Fig. 3A are also questionable? Their data look comparable to Laliberté(2007) in Fig. 3A.

This is an interesting question. The reviewer is correct: the data of Tong et al. (2022) for the binary, aqueous ammonium nitrate agree with the La Liberté (2007) data. We cannot really assign what may have gone wrong with the Tong et al. (2022) experiment for the mixed system; see also the corresponding reply to the same question of reviewer 1. A simple explanation – which we are not able to prove – would be that the OIR was different from the reported one. We decided to delete the last sentence of this paragraph for the revised version.

(4) Line 258, in your system, what is the possible reason for the significant increase in viscosity below $a_w$ of 0.1?

We do see a sudden increase of at least 5 orders of magnitude in viscosity only in the data for the OIR=1 system (cp. Fig. 4) below aw equal 0.1. As written in the manuscript we do not have any visual indication that $NH_4NO_3$ crystallized at this water activity, but we do not have any proof that this did not happen either. We will soften our statement for the revised manuscript by adding the following sentence to the paragraph: "However, we cannot prove that no crystallization of $NH_4NO_3$ occurred, causing the sudden increase in viscosity below aw equal 0.1 for the mixed droplets with OIR=1."

(5) Line 315-316, the authors calculated the diffusion coefficients of $NH_4NO_3$ via the Stokes-Einstein relation. However, previous studies have showed that the Stokes-Einstein relation is not suitable (with large underestimation) to predict the diffusion coefficients for such small diffusing molecules (Evoy et al., 2020; Price et al., 2015). Please also double check the calculation of $D_{H2O}$ (Line 613) in Equation (C1).

As the reviewer writes, calculating diffusivity via Stokes-Einstein for small molecules potentially underestimates the diffusivity for small molecules, see e.g. in addition to the references mentioned by the reviewer also Bastelberger et al. (2017; Atmos. Chem. Phys., 17, 8453–8471). In the context of the atmospheric implications section such

underestimation will make mixing times even faster. This is stated on line 320 of the manuscript already, but we will add a sentence to the revised manuscript to strengthen the statement: "Therefore, the mixing times calculated using these derived diffusivities represent an upper limit." In addition, we will insert a new paragraph after the sentence.

(6) Line 200: "$10^{12}$ Pa" should be "$10^{12}$ Pa s".

Corrected.

(7) Caption of Figure 6: "the viscosities of $NO_3$–sucrose–$H_2O$" should be "$NaNO_3$–sucrose–$H_2O$".

Corrected.

(8) Table A1, what temperature the viscosity parameterization showed in Table A1 is suitable to?

Good point: this information is missing in the manuscript. The parameterizations in Table A1 refer to 293 K; added to manuscript.

(9) The mixing times showed in Fig. B2 are for which compound? The mixing times of different molecules are very different (refer to my major comment 3).

These are estimations for the mixing of the ammonium ion within aqueous toluene–derived SOA containing $NH_4NO_3$ for a dry mass ratio of organic to inorganic equal 2:1. We argue why we are using the ammonium ion radius in lines 315-325.

(10) Table C2, what $d_{max}$ is? Is it the diameter of film?

That refers to the outer diameter of the droplet in the poke-flow experiments to estimate the time needed for equilibration with respect to RH. We will add a footnote to the table to clarify this meaning.